# MECP2 regulates cortical plasticity underlying a learned behaviour in adult female mice

Keerthi Krishnan[1,*], Billy Y.B. Lau[1,*], Gabrielle Ewall[1], Z. Josh Huang[1] & Stephen D. Shea[1]

Neurodevelopmental disorders are marked by inappropriate synaptic connectivity early in life, but how disruption of experience-dependent plasticity contributes to cognitive and behavioural decline in adulthood is unclear. Here we show that pup gathering behaviour and associated auditory cortical plasticity are impaired in female *Mecp2^het* mice, a model of Rett syndrome. In response to learned maternal experience, *Mecp2^het* females exhibited transient changes to cortical inhibitory networks typically associated with limited plasticity. Averting these changes in *Mecp2^het* through genetic or pharmacological manipulations targeting the GABAergic network restored gathering behaviour. We propose that pup gathering learning triggers a transient epoch of inhibitory plasticity in auditory cortex that is dysregulated in *Mecp2^het*. In this window of heightened sensitivity to sensory and social cues, *Mecp2* mutations suppress adult plasticity independently from their effects on early development.

[1] Cold Spring Harbor Laboratory, 1 Bungtown Road, Cold Spring Harbor, New York 11724, USA. * These authors contributed equally to this work. Correspondence and requests for materials should be addressed to K.K. (email: krishnan@utk.edu).

Rett syndrome (RTT) is a neuropsychiatric disorder predominantly caused by mutations in the X-linked gene methyl CpG-binding protein 2 (MECP2)[1]. Males with mutations of their single copy of the gene suffer neonatal encephalopathy and die in infancy[2], and most surviving patients with RTT are females that are heterozygous for Mecp2 mutations. In these females, random X-chromosome inactivation leads to mosaic wild type MECP2 expression and consequently a syndromic phenotype. Patients with RTT achieve early postnatal developmental milestones, but experience an abrupt developmental regression around 6–12 months[3,4]. They typically survive into middle age[5], exhibiting sensory, cognitive and motor deficits throughout life.

MECP2 is broadly expressed in the developing and adult brain[6,7] and is continually required to maintain adult neural function[8–10]. Moreover, restoration of normal MECP2 expression in adult mice improves symptoms[8–10]. These observations establish that MECP2 is necessary to regulate brain function in adulthood. However, the specific function of MECP2 in the mature brain remains unclear, despite its widely studied role in development.

MECP2 regulates neuronal chromatin architecture and gene transcription[11–13] in response to neural activity and experience during postnatal life[14,15]. The known cellular function of MECP2 and the characteristic timing of disease progression raise the possibility that the regulation of neural circuits by MECP2 is increased during specific windows of enhanced sensory and social experience throughout life. We therefore hypothesized that continued disruptions of experience-dependent plasticity in female mice heterozygous for Mecp2 (Mecp2het) hinders learning during adulthood. We tested this hypothesis in adult female Mecp2het mice using pup retrieval, a learned natural maternal behaviour, which is known to induce experience-dependent auditory cortical plasticity[16–18]. First-time mother mice respond to their pups' ultrasonic distress vocalizations by gathering the pups back to the nest, an essential aspect of maternal care[19,20]. Virgin females with no previous maternal experience ('surrogates') can acquire this behaviour when co-housed with a first-time mother and her pups[16]. Single-unit neural recordings show that proficient pup gathering behaviour is correlated with neurophysiological plasticity in the auditory cortex in both surrogates and mothers[16–18].

Here we report that adult Mecp2het surrogates, and surrogates with conditional knockout of Mecp2 in auditory cortex, exhibit impaired pup retrieval behaviour. Maternal experience-triggered changes in GABAergic interneurons occur in wild-type surrogates, but we found that additional changes were observed in Mecp2het surrogates. Specifically, we observed elevated expression of parvalbumin (PV) and perineuronal nets (PNNs). Increases in expression of these markers are associated with the termination or suppression of plasticity in development and adulthood[21–26]. Genetic manipulation of GAD67, the primary synthetic enzyme for GABA, suppressed increases in PV and PNNs and restored gathering in Mecp2het. Furthermore, specific depletion of the PNNs into the auditory cortex also restored efficient pup retrieval behaviour in Mecp2het. Finally, we found that specific knockout of Mecp2 in PV neurons was sufficient to transiently interfere with pup retrieval behaviour. Altogether, our results show that MECP2 regulates experience-dependent plasticity in the adult auditory cortex.

## Results

### Pup gathering behaviour requires auditory cortex.

To assess the efficacy of cortical plasticity underlying pup gathering learning, we devised an assay for gathering behaviour in nulliparous surrogates (Sur). We chose to examine cortical plasticity underlying the acquisition of gathering behaviour in Sur to eliminate the influence of pregnancy. Our intent was not to study maternal behaviour per se or plasticity in mothers, but to use this assay to study the function of MECP2 in adult experience-dependent plasticity in Sur at the neural circuit and behavioural levels.

Assaying the effects of heterozygous deletion of Mecp2 on gathering behaviour presents several advantages. First, the vast majority of patients with RTT are females and heterozygous for mutations of Mecp2 who exhibit mosaic expression of the wild type protein. Thus, female Mecp2het (ref. 27) are a particularly appropriate model of RTT. Second, we can directly relate a natural, learned adult behaviour to specific, experience-dependent changes in the underlying neural circuitry. Third, we can observe effects on adult learning and plasticity that are distinct from developmentally programmed events in the Sur by studying a window of heightened plasticity that is triggered by exposure to a mother and her pups.

Two 7–10 week old matched female littermates (Sur) were co-housed with a first time mother and her pups from late pregnancy until the fifth day following birth (D5) (Fig. 1a). Sur were virgins with no prior exposure to pups. All three adults (the mother and both Sur) were subjected to a retrieval assay (see Materials and Methods) on D0 (day of birth), D3 and D5.

We confirmed the experience-dependent nature of gathering behaviour by comparing performance of maternally-naive WT (NaiveWT) females with that of SurWT on D5. Performance was assessed by computing a normalized measure of latency (latency index, see Methods) and by counting the number of gathering errors (instances of interacting with a pup and failing to gather it to the nest). SurWT performed significantly better than NaiveWT by both measures (Fig. 1b,c) (NaiveWT: $N = 9$ mice; SurWT: $N = 18$ mice; Mann–Whitney, $P = 0.027$), presumably reflecting maternal experience-dependent plasticity.

Several lines of evidence suggest that auditory cortical responses to ultrasonic distress vocalizations facilitate performance of pup gathering behaviour[16–18,28]. We confirmed this by making bilateral excitotoxic (ibotenic acid) lesions of the auditory cortex in wild type mice. Compared with saline-injected mice, mice with lesions exhibited significantly larger latency indices (Saline: $0.20 \pm 0.034$, $N = 6$ mice; Lesion: $0.66 \pm 0.033$, $N = 6$ mice; Mann–Whitney: $P = 0.0022$) and made more errors (Saline: $1.33 \pm 0.95$ errors, $N = 6$ mice; Lesion: $6.64 \pm 0.91$ errors, $N = 6$ mice; Mann–Whitney: $P = 0.015$).

### MECP2 is required for efficient pup gathering behaviour.

Next, we compared the pup gathering performance of SurHet with that of mothers and SurWT. SurWT retrieved pups to the nest with efficiency (as measured by latency index in Fig. 1d,f) and accuracy (as measured by errors in Fig. 1e,g) that were indistinguishable from the mother (Supplementary Movie 1). By contrast, SurHet exhibited dramatic impairment in gathering behaviour, retrieving pups with significantly longer latency and more errors when compared with the SurWT or mothers (Fig. 1d–g). Moreover, this behaviour did not improve with subsequent testing on D3 and D5 (Fig. 1d,e) ($N = 13$–24 mice; Kruskal–Wallis with Bonferroni correction: H values for latency – D0 = 9.4, D3 = 13.05, D5 = 21.68; H values for error – D0 = 26.07, D3 = 26.31, D5 = 24.32; *post-hoc $P < 0.05$). The variability in behaviour in SurHet can be partly explained by the variability in MECP2 expression in the auditory cortex because of random X-chromosome inactivation. Specifically, SurHet with fewer cells expressing MECP2 performed worse in latency and errors than SurHet with more cells expressing MECP2, showing that the range of variability in SurHet behaviour is correlated with

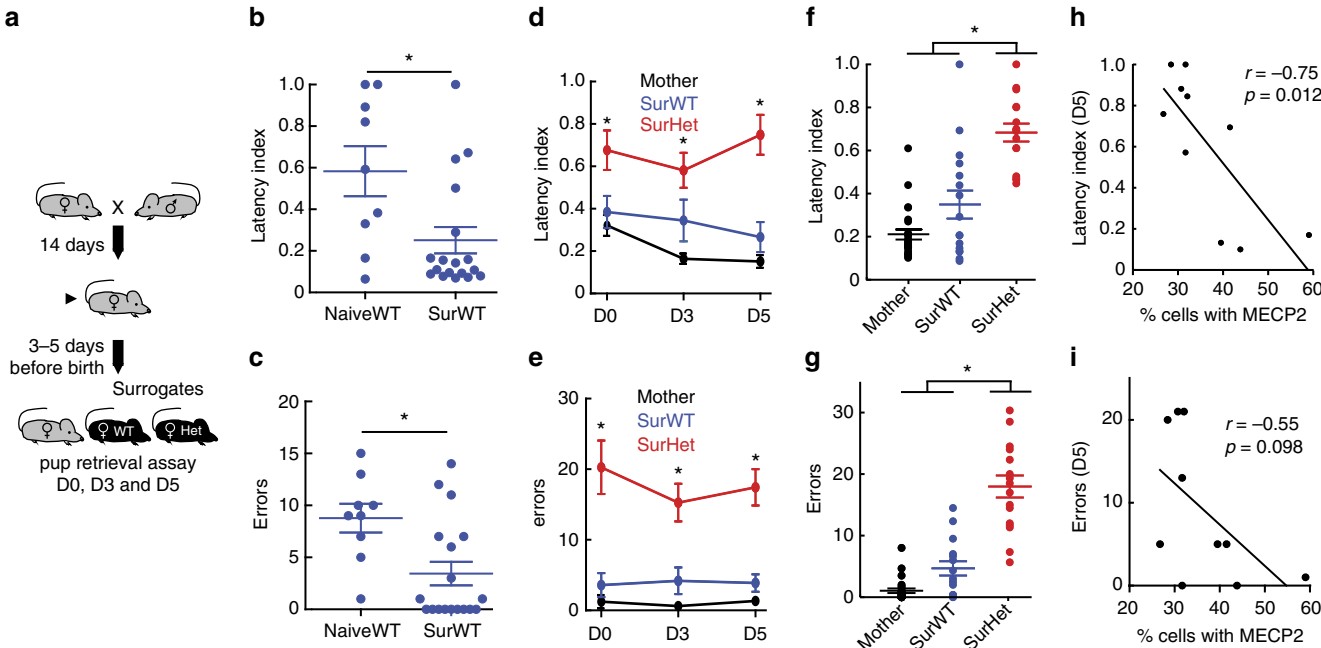

**Figure 1 | Female *Mecp2^het* mice perform poorly at pup retrieval behaviour.** (**a**) Schematic of behavioural paradigm. Virgin *Mecp2^het* (Het) and wild type littermates (WT) mice were co-housed with a pregnant female before birth of pups. Surrogates (Sur) were tested on the pup retrieval task on days 0 (D0), 3 and 5 after birth. (**b,c**) SurWT tested on D5 ($N = 18$ mice) showed significant improvements on a normalized measure of latency to gather (**b**) and reduced number of gathering errors (**c**) compared with pup-naive mice ($N = 9$ mice) (Mann–Whitney, *$P = 0.027$). Lines represent mean ± s.e.m. (**d,e**) Mean performance at D0, D3 and D5 for mothers, SurWT and SurHet as measured by normalized latency (**d**) and errors (**e**). Lines represent mean ± s.e.m. SurHet showed consistently poorer pup retrieval performance than mothers and SurWT in all three sessions ($N = 13$-24 mice; Kruskal–Wallis with Bonferroni correction: H values for latency — D0 = 9.4, D3 = 13.05, D5 = 21.68; H values for error—D0 = 26.07, D3 = 26.31, D5 = 24.32; *$P < 0.05$). (**f,g**) Mean performance of normalized latency (**f**) and errors (**g**) averaged over all three sessions ($N = 13$-24 mice; Kruskal-Wallis with Bonferroni correction: latency—H = 29.95, error—H = 35.45; **post-hoc* $P < 0.05$. SurHet had significantly longer latency and made more errors compared with mothers and SurWT. Mean ± s.e.m. are shown in line. (**h,i**) At D5 Sur, Het performance of normalized latency (**h**) and errors (**i**) negatively correlated with percentage of cell population expressing MECP2 ($N = 10$ mice; Pearson's r: For H: r = − 0.75, $P = 0.0012$; for I: r = − 0.55, $P = 0.098$).

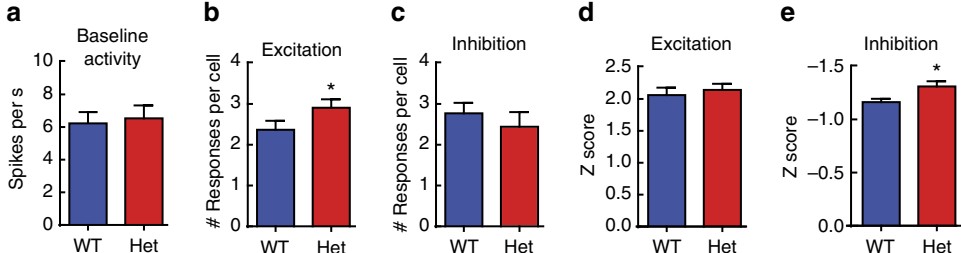

**Figure 2 | Auditory cortex activity is grossly similar in NaiveHet and NaiveWT.** (**a**) Baseline spontaneous activity was not different between NaiveWT and NaiveHet (WT: $n = 99$ cells, 11 mice; Het: $n = 87$ cells, 13 mice; Mann–Whitney, $P = 0.70$). (**b,c**) NaiveHet neurons were excited by a small but significantly greater number of stimuli (**b**; WT: $n = 56$ cells, 11 mice; Het: $n = 66$ cells, 13 mice; Mann-Whitney, *$P = 0.047$), but inhibited by a similar number of stimuli compared with NaiveWT (**c**; WT: $n = 47$ cells, 11 mice; Het: $n = 24$ cells, 13 mice; Mann–Whitney, $P = 0.33$). (**d,e**) Response strength, measured as a z score, was not significantly different between NaiveWT and NaiveHet, for excitation (**d**) but was significantly increased in NaiveHet for inhibition (**e**) (Excitation: WT: $n = 136$ responses, 56 cells, 11 mice; Het: $n = 192$ responses, 66 cells, 13 mice; Mann–Whitney, $P = 0.43$; Inhibition: WT: $n = 133$ responses, 47 cells, 11 mice; Het: $n = 59$ responses, 24 cells, 13 mice; Mann–Whitney, *$P = 0.0054$). (**a**–**e**) Bar graphs represent mean ± s.e.m.

MECP2 expression in the auditory cortex (Fig. 1h,i) ($N = 10$ mice; Pearson's r). Taken together, the results demonstrate that MECP2 is required for successful acquisition of this learned behaviour.

In these experiments, we used a germline *Mecp2* knockout that affects MECP2 expression throughout the animal. Therefore, the poor pup gathering performance of SurHet could, in principle, be because of motor deficits or deafness. We found no significant difference in movement during behavioural trials between the genotypes (SurWT: 2,059 ± 216.5 significant motion pixels (SMP), $N = 8$ mice; SurHet: 2,139 ± 259.9 SMP, $N = 8$ mice;

Mann–Whitney: $P = 0.78$), consistent with previous findings that *Mecp2^het* lack robust motor impairments[29].

We also found no evidence that *Mecp2^het* are deaf or otherwise insensitive to sound, consistent with a previous study[30]. Neurons in the auditory cortex of NaiveHet exhibited widespread and robust responses to auditory stimuli. Baseline spontaneous activity was comparable between NaiveWT and NaiveHet (Fig. 2a) (WT: $n = 99$ cells, 11 mice; Het: $n = 87$ cells, 13 mice; Mann–Whitney, $P = 0.70$). Analysis of stimulus-evoked responses showed that auditory cortex neurons of NaiveHet showed a slight

increase in excitatory responses to a larger number of stimuli compare to NaiveWT (Fig. 2b; WT: $n = 56$ cells, 11 mice; Het: $n = 66$ cells, 13 mice; Mann–Whitney, $*P = 0.047$), but did not show differences in the number of inhibitory responses (Fig. 2c; WT: $n = 47$ cells, 11 mice; Het: $n = 24$ cells, 13 mice; Mann–Whitney, $P = 0.33$). Moreover, the response strengths for excitation were comparable between NaiveWT and NaiveHet (Fig. 2d; WT: $n = 136$ responses, 56 cells, 11 mice; Het: $n = 192$ responses, 66 cells, 13 mice; Mann–Whitney, $P = 0.43$), whereas the response strengths for inhibition were slightly increased in NaiveHet compared with NaiveWT (Fig. 2e; WT: $n = 133$ responses, 47 cells, 11 mice; Het: $n = 59$ responses, 24 cells, 13 mice; Mann–Whitney, $*P = 0.0054$). Taken together, these data establish that the impaired pup gathering behaviour in $Mecp2^{het}$ is not caused by frank deafness or insensitivity of the auditory system in naive females.

**MECP2 in adult auditory cortex is required for pup gathering.** Measuring behavioural effects in germline mutants leaves open the possibility of a requirement for MECP2 in early postnatal development and/or in other brain regions. Therefore, we used a conditional deletion approach to specifically deplete MECP2 expression in the auditory cortex by bilaterally injecting AAV-GFP-Cre (adeno-associated virus expressing CRE recombinase) in 4-week old $Mecp2^{flox/flox}$ mice[31] (Fig. 3a). Histological analysis of sections from SurMecp2$^{flox/flox}$ five weeks after injection with AAV-Cre showed >91% of GFP expressing (GFP$^+$) nuclei in the auditory cortex ($n = 685$ GFP$^+$ cells, 12 images, 3 mice) (see methods) were devoid of MECP2 expression (Fig. 3b–f). We counted non-GFP expressing (GFP$^-$) and GFP$^+$ cells to determine the extent of MECP2 knock-down in the GFP$^+$ cells and found significant reduction of MECP2 expression in the GFP$^+$ cells of the auditory cortex (Fig. 3f; $n = 119$ cells per cell type, 3 mice; Mann–Whitney, $*P < 0.05$).

$Mecp2^{flox/flox}$ mice injected with AAV-GFP alone (control), consistently showed strong pup gathering performance (Fig. 3g,h) ($N = 14$ mice). In contrast, $Mecp2^{flox/flox}$ mice injected with AAV-Cre exhibited variable pup gathering behaviour that depended on the proportion of auditory cortex affected by the injection. The degree of impairment for an individual mouse was positively correlated with the percentage of the auditory cortices encompassed by the virus injection site (Fig. 3g,h) (Latency: $r = 0.80$, $P = 0.0006$; $N = 14$ mice; Errors: $r = 0.83$, $P = 0.0002$; $N = 14$ mice; Pearson's $r$). No positive correlation between injection area and behavioural performance was found with regions surrounding the auditory cortex (Latency: $r = 0.40$, $P = 0.16$; Errors: $r = 0.25$, $P = 0.40$; $N = 14$ mice; Pearson's $r$). Taken together, these findings demonstrate that MECP2 expression, specifically in the auditory cortex of mature females, is required for proficient learning of pup gathering behaviour.

**SurHet exhibit altered plasticity of GABAergic interneurons.** The regional requirement for MECP2 led us to examine maternal experience-dependent molecular events in the auditory cortex. Recent data on the neurophysiological correlates of maternal learning suggest that there are changes in inhibitory responses of vocalizations in the auditory cortex of mothers and surrogates[17,32]. There is also evidence that inhibitory networks are particularly vulnerable to $Mecp2$ mutation[33–35]. For these reasons, we focused our attention on experience-dependent dynamics of molecular markers associated with inhibitory circuits.

We used immunostaining of brain sections from the auditory cortex of Sur and naive females to examine experience-induced molecular events in inhibitory networks of $Mecp2^{het}$ and

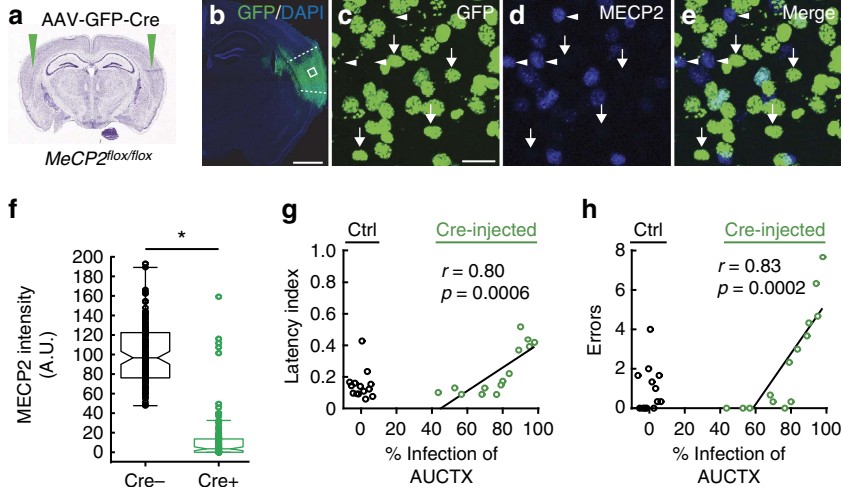

**Figure 3 | MECP2 expression in the auditory cortex is required for efficient pup retrieval.** (**a**) Diagram depicting AAV-GFP-Cre injection into the auditory cortex (green arrows) of female $Mecp2^{flox/flox}$ mouse. These mice also carried a nuclear localized and Cre-dependent GFP allele (H2B-GFP) that allowed us to directly visualize Cre-positive cells. (**b**) Photomicrograph of a brain section from a $Mecp2^{flox/flox}$ mouse with AAV-GFP-Cre injection and counterstained with the nuclear marker, DAPI. Dotted lines mark the boundary of auditory cortex. Scale bar = 1 mm. (**c–e**), Magnified confocal images of a selected region boxed in B. GFP$^+$ cells (**c**, green) are negative for MECP2, as confirmed by anti-MECP2 immunostaining (**d** and **e**, blue) (91.2 ± 0.03%; $n = 685$ GFP$^+$ cells, 12 images, 3 mice). Arrows point to GFP$^+$ cells that are MECP2$^-$. Arrowheads point to GFP$^-$ cells that are MECP2$^+$, which served as a positive control for MECP2 staining. Scale bar, 20 µm, applies to **c–e**. (**f**) Mean MECP2 expression (intensity; A.U. = arbitrary units) in AAV-GFP-Cre infected cells (GFP$^+$) and uninfected cells (GFP$^-$) in the same AAV-GFP-Cre injected animals ($n = 119$ cells per cell type, 3 mice; Mann–Whitney, $*P < 0.05$). Cre-infected cells showed significantly reduced MECP2 expression compared with uninfected cells. Boxplot with standard Matlab-generated whiskers are shown. Notches represent 95% confidence interval of median. Each dot overlaid on the boxplot represents a cell. (**g,h**) Correlation analysis showed a significant positive relationship between the proportion of auditory cortex expressing GFP-Cre and both gathering latency (**g**) and number of errors (**h**) (green dots; $N = 14$ mice; Pearson's $r$: For G: $r = 0.80$, $P = 0.0006$; for H: $r = 0.83$, $P = 0.0002$). Control $Mecp2^{flox/flox}$ mice injected with AAV-GFP alone (ctrl; black dots) showed normal behaviour ($N = 14$ mice).

wild-type littermates. Expression of GAD67, the key rate-limiting enzyme for GABA synthesis, was significantly increased five days after initiation of maternal experience in mutant and wild type mice (Fig. 4a,b) ($n = 36$–451 cells, 12–32 images, 4–8 mice; ANOVA: Tukey's *post-hoc* test, $*P < 0.05$). For both Sur genotypes, expression returned to baseline by the time the pups were weaned (D21) (Fig. 4a,b). This suggests that maternal experience triggers transient experience-dependent molecular changes in inhibitory neurons in the auditory cortex of Sur mice.

In SurHet only, we observed transient increases in additional markers of inhibitory networks that are often associated with suppressing plasticity. For example, recent work has linked high parvalbumin (PV)-expressing inhibitory networks to reduced capacity for adult learning and plasticity[25], and the closure of developmental critical periods[21,24,36]. We detected a maternal experience-induced shift in the intensity distribution of PV immunofluorescence in SurHet but not SurWT (Fig. 5a,b,f). The intensity distribution for SurHet was fit with a mixture of two Gaussians to define high and low PV-expressing populations. The proportion of high PV-expressing neurons was significantly greater in SurHet than in any other group (Fig. 5b) ($n = 2704$–4906 cells, 19–20 images, 5 mice; ANOVA: Tukey's *post-hoc* test, $*P < 0.05$ compared with all other groups).

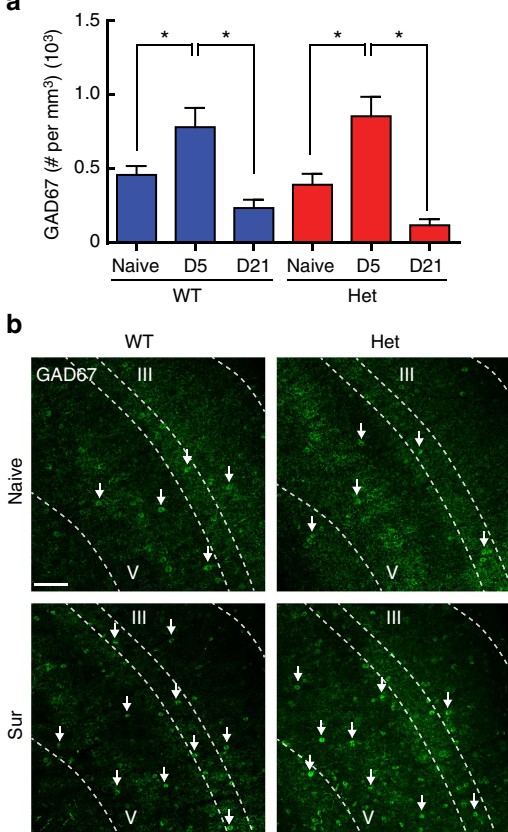

**Figure 4 | Maternal experience transiently enhances GAD67 expression level in the auditory cortex of wild-type and *Mecp2het* mice. (a)** The density of high-intensity GAD67 cells was significantly increased in both SurWT (dark blue) and SurHet (red) at D5, and returned to naive levels at D21 ($n = 36$–451 cells, 12–32 images, 4–8 mice; ANOVA: Tukey's *post-hoc* test, $*P < 0.05$). Bar graphs represent mean ± s.e.m. **(b)** Representative confocal images taken from the auditory cortex of a NaiveWT and NaiveHet (top row) and SurWT and SurHet at D5 (bottom row). Arrows point to high-intensity GAD67 cells. Scale bar, 100 μm, applies to all images. Dashed lines delineate cortical layers with layers III and V indicated.

Mature neural circuits are often stabilized by perineuronal nets (PNNs), which are composed of extracellular matrix proteins such as chondroitin sulfate proteoglycans[37], and mainly surround PV$^+$ GABAergic interneurons in the cortex[38]. We observed a dramatic experience-dependent increase in the number of high-intensity PNNs in SurHet but not in SurWT (Fig. 5c,g) ($n = 292$–1,735 PNN$^+$ cells, 12–38 images, 3–9 mice; ANOVA: Tukey's *post-hoc* test, $*P < 0.05$ compared with all other groups). Importantly, both PV and PNNs returned to baseline levels in surrogates by weaning age of the pups (D21) (Fig. 5b,c). In addition, the percentage of PNN that co-localized with PV$^+$ cells was unchanged among all groups of mice (Fig. 5d,h) ($n = 1$–107 PNN$^+$ cells, 1–103 PV cells, 6 images, 3 mice; ANOVA: Tukey's *post-hoc* test, $P > 0.05$). However, SurHet at D5 showed an increased percentage of PV$^+$ cells co-localized with PNN (Fig. 5e) ($n = 92$–319 PV cells, 1–103 PNN$^+$ cells, 6 images, 3 mice; ANOVA: Tukey's *post-hoc* test, $*P < 0.05$ compared with all other groups except SurWT P21). Thus, maternal experience triggers temporally-restricted changes to inhibitory circuits in SurHet, but there are also additional changes not observed in SurWT, including elevated PV and PNN expression. We have separately observed elevated PV and PNN expression and altered plasticity in the visual cortex of *Mecp2*-null males during the visual critical period[39]. Similar changes may act to limit network plasticity after maternal experience. Moreover, the reversion to baseline levels following weaning indicates that pathological features of the plasticity are temporally limited and suggests that certain aspects of *Mecp2het* pathology are only revealed during appropriate experiences that occur within that window.

**Rescue of SurHet phenotypes by *Gad1* manipulation.** GAD67 is an activity-regulated, rate-limiting enzyme that synthesizes the cortical inhibitory neurotransmitter GABA. GAD67 expression levels also correlate highly with PV levels[25] and regulate PV neuron maturation[40]. Several recent studies suggest that mice that are heterozygous for loss of the GAD67 gene (*Gad1*) exhibit lower levels of PV expression[41,42]. We have separately observed that lowering GAD67 levels in the *Mecp2*-null male mice normalized expression of PV and PNN in the developing visual cortex[39]. We therefore speculated that genetically manipulating GAD67 expression (*Gad1het*) in *Mecp2het* might result in normalization of PV network-associated markers in the adult auditory cortex. To test this idea, we crossed germline *Gad1het* mice into the *Mecp2het* background and examined the effects on maternal experience-dependent changes in PV and PNNs.

As expected, naive WT and *Mecp2het* carrying the *Gad1het* allele (NaiveWT;Gad1$^{het}$ and NaiveHet;Gad1$^{het}$, respectively) showed half the GAD67 expression seen in WT and *Mecp2het* (NaiveWT: 458.9 ± 60.6 cells per mm$^3$, NaiveHet: 393.5 ± 73.3 cells per mm$^3$, NaiveWT;Gad1$^{het}$: 174.6 ± 60.9 cells per mm$^3$, NaiveHet;Gad1$^{het}$: 193.2 ± 41.3 cells per mm$^3$; $n = 92$–334 cells, 20–32 images, 5–8 mice; T-test: $P < 0.05$ NaiveHet;Gad1$^{het}$ compared with NaiveWT and NaiveHet; T-test: $P < 0.05$ NaiveWT;Gad1$^{het}$ compared with NaiveWT and NaiveHet). In contrast to SurHet, SurHet;Gad1$^{het}$ exhibited a correction in the maternal experience-dependent increase in PV expression levels (Fig. 6a,b) and had a significantly lower proportion of high-intensity PV$^+$ cells (Fig. 6b) ($n = 4,353$–5,079 cells, 16–20 images, 4–5 mice; ANOVA: Tukey's *post-hoc* test, $*P = 0.02$). We also saw significantly fewer PNNs in the double mutants (Fig. 6d) ($n = 196$–1,735 PNN$^+$ cells, 17–38 images, 4–9 mice; ANOVA: Tukey's *post-hoc* test, $*P = 0.01$). NaiveWT;Gad1$^{het}$ exhibited a significantly elevated percentage of high-intensity PV$^+$ cells, compared with NaiveWT (Fig. 6c), likely because of compensatory effects of long-term genetic reduction of GAD67.

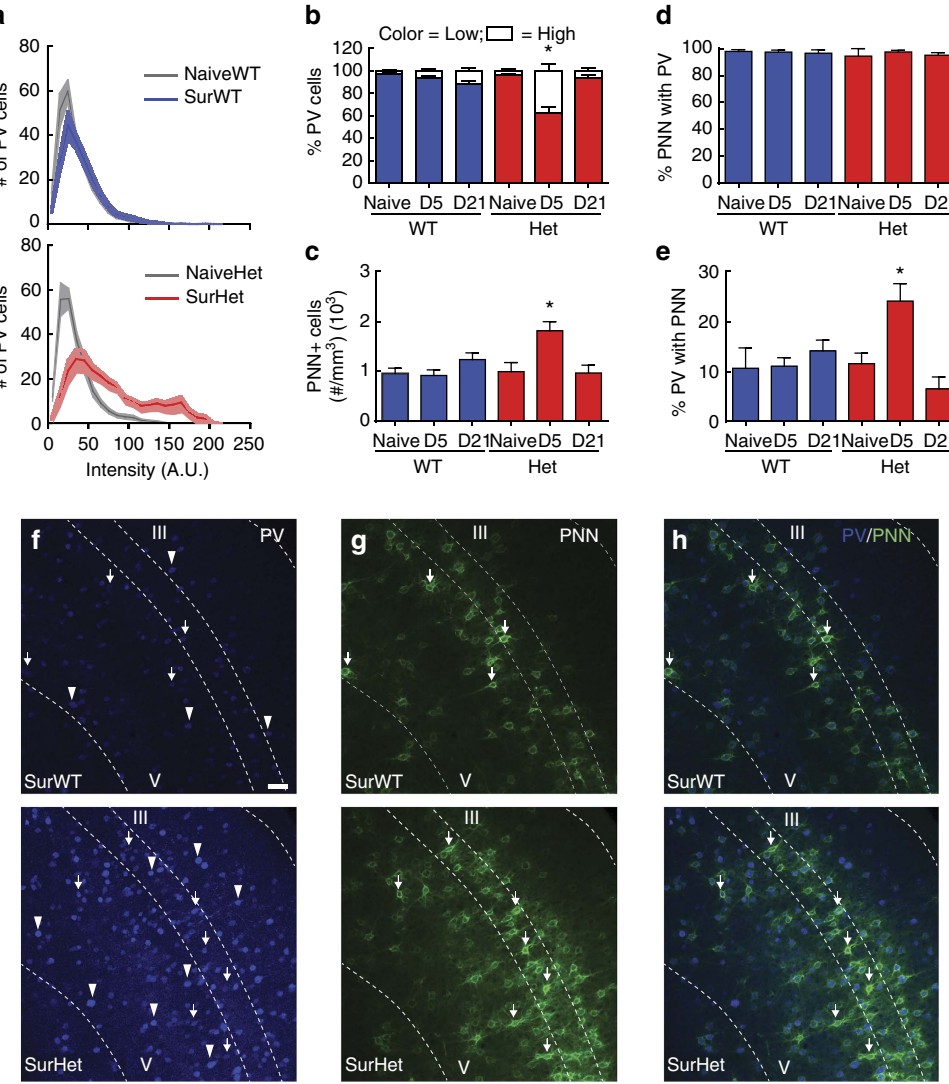

**Figure 5 | Female *Mecp2^het* mice exhibit abnormal maternal experience-induced changes to inhibitory networks in the auditory cortex.** (**a**) Histograms showing the mean distribution of PV cell intensity in adult surrogates 5 days after pup exposure (D5). Top panel, distribution of PV cell intensity is similar between SurWT (dark blue) and NaiveWT (grey). Bottom panel, there is a shift in the distribution toward elevated PV expression in SurHet (red) compared with NaiveHet (grey) ($n = 2704$–$4906$ PV$^+$ cells, 19–20 images, 5 mice for each group). The solid line and shaded region represent mean ± s.e.m. respectively, in both panels. (**b**) The shift reflects a significant transient increase in high-PV expressing cells at D5 that returned to baseline at D21 in SurHet (ANOVA: Tukey's *post-hoc* test, *$P < 0.05$ compared with all other groups). (**c**) The density of high-intensity perineuronal nets (PNNs) was significantly increased only in SurHet at D5 ($n = 292$–$1735$ PNN$^+$ cells, 12–38 images, 3–9 mice; ANOVA: Tukey's *post-hoc* test, *$P < 0.05$ compared with all other groups), and returned to baseline at D21. (**d**) The percentage of PNN co-localizing with PV-expressing cells was not significantly different across genotypes and conditions ($n = 1$–$107$ PNN$^+$ cells, 1–103 PV cells, 6 images, 3 mice; ANOVA: Tukey's *post-hoc* test, $P > 0.05$). (**e**) However, the percentage of PV cells co-localizing with PNN was significantly higher in SurHet at D5 ($n = 92$–$319$ PV cells, 1–103 PNN$^+$ cells, 6 images, 3 mice; ANOVA: Tukey's *post-hoc* test, *$P < 0.05$ compared with all other groups except SurWT P21). (**b**–**e**) Bar graphs represent mean ± s.e.m. (**f**–**h**) Representative confocal images taken from the auditory cortex of a SurWT and SurHet showing relative expression of PV (**f**) and PNN (**g**). Arrowheads indicate high-intensity PV cells. Arrows point to co-localization of PV and PNN. Scale bar, 50 μm, applies to all images. Dashed lines delineate cortical layers with layers III and V indicated.

Interestingly, this increase was not seen after maternal experience (Fig. 6c), returning to the appropriate activity-dependent expression of PV that was not significantly different from the WT ($n = 3,561$–$4,782$ cells, 16–20 images, 4–5 mice; ANOVA: Tukey's *post-hoc* test, *$P < 0.05$). There was no change in PNNs in this genotype, before or after maternal experience (Fig. 6e) ($n = 319$–$780$ PNN$^+$ cells, 16–28 images, 4–7 mice; ANOVA: Tukey's *post-hoc* test, $P > 0.05$). These data indicate that manipulating GAD67 in the *Mecp2*-deficient background ameliorates features of impaired maternal experience-dependent auditory cortical plasticity in SurHet.

We next assessed whether the corrective effect of GAD67 reduction on inhibitory markers in SurHet reinstated learning. Remarkably, SurHet;Gad1^het exhibited significant decreases in latency index (Fig. 6f) and the number of errors (Fig. 6g) (SurHet;Gad1^het: $N = 7$ mice; SurWT: $N = 18$ mice; SurHet: $N = 18$ mice; ANOVA: Tukey's *post-hoc* test, *$P < 0.05$) when compared with SurHet. In fact, the gathering performance of SurHet;Gad1^het was indistinguishable from that of SurWT or SurWT;Gad1^het (Fig. 6f,g) (SurWT;Gad1^het: $N = 7$ mice). These results show that manipulating GABAergic neurons in the *Mecp2*-deficient background alleviates

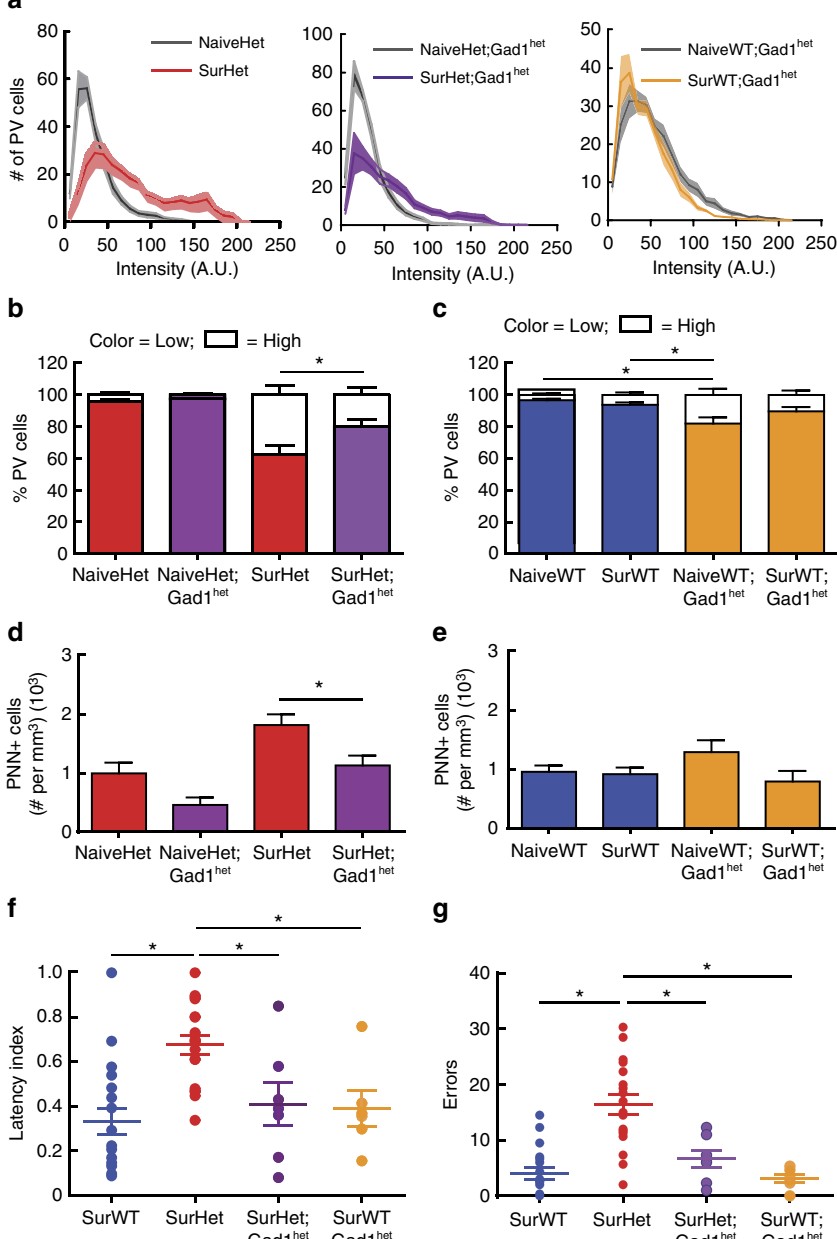

**Figure 6 | Genetic manipulation of the GABA synthesizing enzyme *Gad1* rescues cellular and behavioural phenotypes in *Mecp2het*.** (**a**) Histograms showing the mean distribution of PV cell intensity comparing SurHet (left, red), SurHet;Gad1het (middle, purple) and SurWT;Gad1het (right, orange) at D5 to their respective naive genotypes (grey). SurHet;Gad1het showed a smaller shift toward elevated PV expression after maternal experience ($n = 4353$–$5079$ PV$^+$ cells, 16–20 images, 4–5 mice). The solid line and shaded region represent mean ± s.e.m., respectively. (**b**) SurHet;Gad1het showed a significant decrease in the high-intensity PV population compared with SurHet at D5 (ANOVA: Tukey's *post-hoc* test, *$P = 0.02$). (**c**) NaiveWT;Gad1het showed significantly more high-intensity PV cells compared with NaiveWT and SurWT. Upon maternal experience, the PV population of SurWT;Gad1het shifted towards WT PV expression levels ($n = 3561$–$4782$ PV$^+$ cells, 16–20 images, 4–5 mice; ANOVA: Tukey's *post-hoc* test, *$P < 0.05$). (**d**) At D5, high-intensity PNN densities were significantly reduced in SurHet;Gad1het, compared with SurHet ($n = 196$–$1735$ PNN$^+$ cells, 17–38 images, 4–9 mice; ANOVA: Tukey's *post-hoc* test, *$P = 0.01$). (**e**) High-intensity PNN densities were not significantly different between WT; Gad1het and WT mice, before and 5 days after maternal experience ($n = 319$–$780$ PNN$^+$ cells, 16–28 images, 4–7 mice; ANOVA: Tukey's *post-hoc* test, $P > 0.05$). (**b**–**e**) Bar graphs represent mean ± s.e.m. (**f,g**) Pup retrieval behaviour is significantly improved in SurHet; Gad1het (purple) ($N = 7$ mice) as measured by normalized latency (**f**) and errors (**g**) averaged across three sessions (SurWT: $N = 18$ mice; SurHet: $N = 18$ mice; SurWT; Gad1het: $N = 7$ mice. ANOVA: Tukey's *post-hoc* test, *$P < 0.05$). Mean ± s.e.m. are shown.

learning deficits, potentially through effects on levels of PV and PNNs.

**Suppressing PNN formation of SurHet improves pup gathering.** PNNs are known to act as barriers to structural plasticity[23,24].

Thus, relief from the excessive formation of PNNs in SurHet;Gad1het could be a critical factor allowing efficient pup gathering. We speculated that suppressing PNN formation selectively in the auditory cortex just before maternal experience is sufficient to improve behavioural performance of

SurHet. We therefore made bilateral auditory cortical injections of chondroitinase ABC (ChABC), which dissolves and suppresses the formation of PNNs (ref. 24), thereby allowing for the formation of new synaptic contacts[37]. Two sites of injection were made into each hemisphere one to three days before initiating assessment of retrieval performance (see Materials and Methods). Injection of ChABC into the auditory cortex of Het and WT significantly reduced high-intensity PNN counts compared with their respective controls: penicillinase-injected[24] mice (Fig. 7a–d) (Het-Pen: $n = 710$ PNN$^+$ cells, 31 images, 8 mice; Het-ChABC: $n = 273$ PNN$^+$ cells, 24 images, 6 mice; Mann Whitney, *$P = 0.0003$; WT-Pen: $n = 455$ PNN$^+$ cells, 32 images, 8 mice; WT-ChABC: $n = 108$ PNN$^+$ cells, 32 images, 8 mice; Mann Whitney: *$P < 0.0001$). SurHet mice that received bilateral injections of ChABC in the auditory cortex showed significantly improved gathering performance of D5 pups. ChABC-injected SurHet retrieved pups with lower latency index (Fig. 7e,g) and fewer errors (Fig. 7f,h) compared with SurHet injected with the control enzyme, penicillinase (at D5: Het-Pen: grey line, $N = 12$ mice; Het-ChABC: red line, $N = 10$ mice; Mann–Whitney, *$P < 0.05$). ChABC-injected WT performed similarly to the penicillinase-injected WT, with a small significant decrease in latency index at Day 3 (Fig. 7e–h) (For clarity, only WT-ChABC data are shown in the blue line, $N = 5$ mice; WT-Pen: normalized latency index – D0 = $0.43 \pm 0.13$, D3 = $0.29 \pm 0.12$, D5 = $0.19 \pm 0.07$, $N = 7$ mice, at D3: Mann–Whitney, $P = 0.048$; WT-Pen: errors – D0 = $2.3 \pm 1.4$ errors, D3 = $1.6 \pm 0.78$ errors, D5 = $1.57 \pm 0.81$ errors, $N = 7$ mice; Mann–Whitney, $P > 0.05$).

Not all injections covered the entire auditory cortex, because of technical issues. Hence, we correlated the percentage of the region affected by the injection with gathering performance. In SurHet, the proportion of auditory cortex bilaterally encompassed by the injection site was significantly negatively correlated with latency index (Fig. 7i) ($N = 13$ mice, $r = -0.75$, $P = 0.0033$, Pearson's $r$) and number of errors (Fig. 7j) ($N = 13$ mice, $r = -0.75$, $P = 0.0034$, Pearson's $r$) exhibited on D5 pups. Interestingly, this relationship did not emerge until day 5 of maternal experience. Therefore, increased PNNs in SurHet inhibit auditory cortical plasticity that is required for rapid and accurate pup gathering.

**Knocking out *Mecp2* in PV neurons affects early learning.** Lack of MECP2 expression in PV$^+$ neurons contributes to distinct RTT-like phenotypes[35] and affects critical period plasticity in the visual cortex[43]. To determine the role of MECP2 in PV neurons in the pup retrieval behaviour, we crossed *Mecp2$^{flox}$* (ref. 31) mice with *PV-Cre* mice[44]. *Mecp2$^{flox}$/PV$^{cre}$* (PV-KO) mice displayed significant impairment in latency and errors on D0, but improved significantly to WT performance by D3 and D5 (Fig. 8a–d; Normalized Latency: CTRL: $N = 11$ mice; PV-KO: $N = 9$ mice; Mann–Whitney, *$P = 0.020$; Errors: CTRL: $N = 11$ mice; PV-KO: $N = 9$ mice; Mann–Whitney, *$P = 0.010$). In agreement with the behaviour results, PNN numbers were similar between WT and PV-K0 at D5 (Fig. 8e; CTRL: $n = 514$ PNN$^+$ cells, 36 images, 9 mice; PV-KO: $n = 344$ PNN$^+$ cells, 34 images, 9 mice; Mann–Whitney, $P = 0.064$). These results potentially reveal a

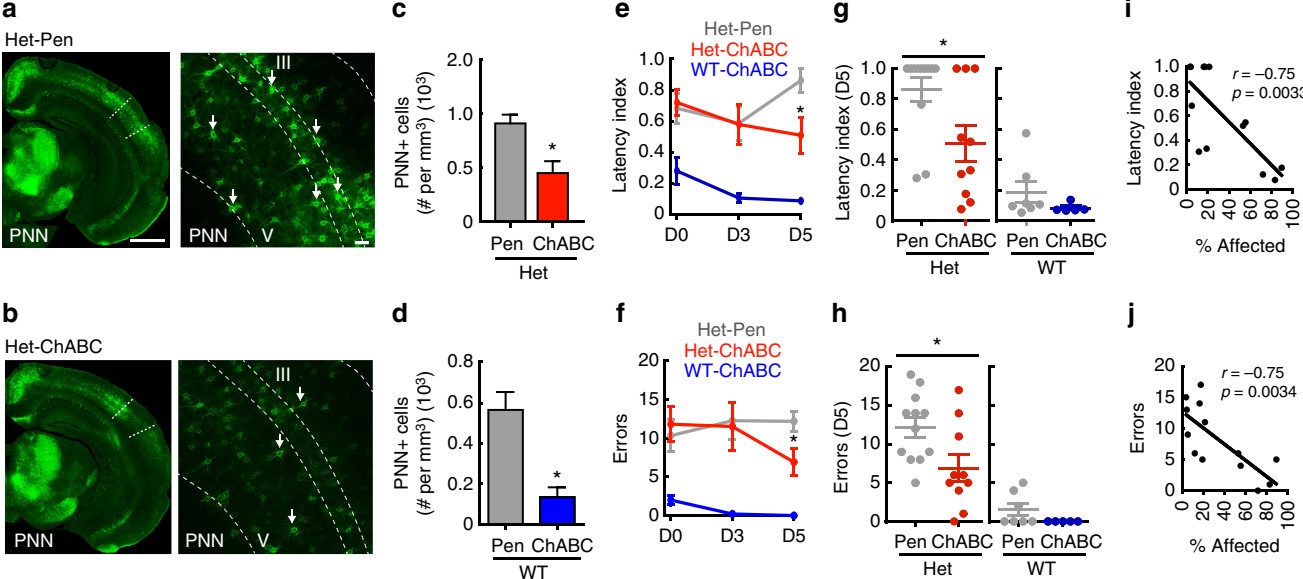

**Figure 7 | Pharmacological suppression of PNN formation in the auditory cortex restores wild-type behaviour in *Mecp2$^{het}$*.** (**a,b**) Samples of low (left panel) and high (right panel) magnification confocal images taken at D5, from the auditory cortex of SurHet that received injections of either a control enzyme (**a**, penicillinase) or an enzyme that dissolves PNNs (**b**, chondroitinase ABC). Arrows indicate high-intensity PNNs. Dashed lines delineate cortical layers with layers III and V indicated. Scale bars in **a**: left image = 1 mm, right = 50 μm, which also apply to the respective images in **b**. (**c,d**) At D5, chondroitinase ABC (ChABC) significantly dissolved PNNs in the injected brains of Het (**c**) and WT (**d**) compared with their respective penicillinase (Pen) -injected genotypes (Het-Pen: $n = 710$ PNN$^+$ cells, 31 images, 8 mice; Het-ChABC: $n = 273$ PNN$^+$ cells, 24 images, 6 mice; Mann Whitney, *$P = 0.0003$; WT-Pen: $n = 455$ PNN$^+$ cells, 32 images, 8 mice; WT-ChABC: $n = 108$ PNN$^+$ cells, 32 images, 8 mice; Mann Whitney: *$P < 0.0001$). Bar graphs represent mean ± s.e.m. (**e–h**) Pup retrieval behaviour improved significantly on D5 in SurHet injected with ChABC (orange), as measured by normalized latency (**e,g**) and errors (**f,h**) compared with penicillinase-injected SurHet (grey) (Het-Pen: $N = 12$ mice; Het-ChABC: $N = 10$ mice; ANOVA: Tukey's *post-hoc* test, *$P < 0.05$). No significant differences in latency and errors were observed between ChABC-injected and penicillinase-injected WT except at D3 (Mann–Whitney, $P = 0.048$). For simpler graphic presentation, only ChABC-injected WT data are shown in blue (WT-Pen: $N = 7$ mice; WT-ChABC: $N = 5$ mice). Mean ± s.e.m. are shown. (**i,j**) Correlation analysis showed a significant negative relationship between the proportion of auditory cortex encompassed by chondroitinase ABC injection for both latency (**i**) and number of errors (**j**) at Day 5 ($N = 13$ mice; Pearson's $r$: For I: $r = -0.75$, $P = 0.0033$; for J: $r = -0.75$, $P = 0.0034$).

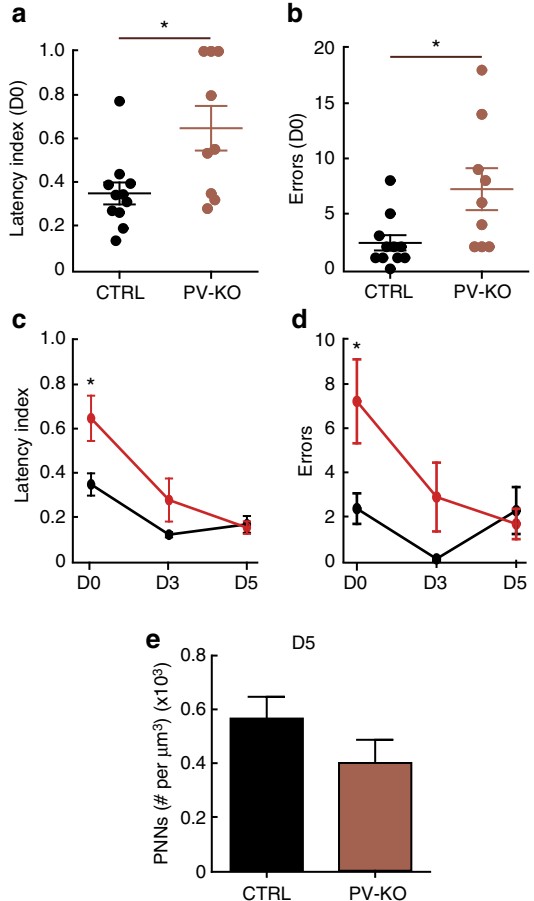

**Figure 8 | Knocking out *Mecp2* in PV neurons affects early learning.**
(**a**–**d**), Mice with PV cells lacking MECP2 (PV-KO) behaved significantly worse than their control littermates (CTRL) at Day 0 (D0) by measure of latency (**a**,**c**) and errors (**b**,**d**) (CTRL: $N = 11$ mice; PV-KO: $N = 9$ mice; At D0: Latency: Mann–Whitney, *$P = 0.020$; errors: Mann–Whitney, *$P = 0.010$). However, PV-KO mice behaved equally as well as their control littermates at Day 3 and 5 (D3 and D5, respectively (**c**,**d**; Mann–Whitney, $P > 0.05$). Lines represent mean ± s.e.m. (**e**) Density of high-intensity PNN$^+$ cells were comparable between PV-KO mice and their control littermates accessed at Day 5 (D5) (CTRL: $n = 514$ PNN$^+$ cells, 36 images, 9 mice; PV-KO: $n = 344$ PNN$^+$ cells, 34 images, 9 mice; Mann–Whitney, $P = 0.064$). Bar graphs represent mean ± s.e.m.

dynamic role for MECP2 in PV neurons during pup retrieval behaviour. Further work will be required to define the time course and molecular mechanisms mediating the change in plasticity between D0 and D3.

## Discussion

A key challenge for understanding the pathogenesis of RTT and neuropsychiatric disorders in general is to identify the associated molecular and cellular changes and trace the resulting circuit alterations that underlie behaviour deficits. It is also critical to differentiate between impairment of developmental programs and effects on experience-dependent neural plasticity. Here we take advantage of a robust natural behaviour in female mice that relies on a known cortical region, and link molecular events in that region and behaviour. Our data identify a specific critical role for MECP2 in experience-dependent plasticity of cortical inhibitory networks in adults.

Most previous studies in mouse models of RTT were conducted in *Mecp2*-null male mice, because they exhibit earlier

and more severe phenotypes in many assays. Therefore, with the exception of a few studies[29,45,46], the molecular, circuit and behavioural defects in *Mecp2*^*het* female mice are largely unknown. Since RTT affects more females, *Mecp2*^*het* female mice represent a more translationally-relevant model of RTT than *Mecp2*-null male mice.

We found a robust behavioural phenotype in the *Mecp2*^*het* mice, suggesting impairment of adult experience-dependent plasticity. We conclude that dysregulated auditory processing in the cortex, because of impaired inhibitory neuronal plasticity, leads to altered learned behaviour. We also showed that when normal plasticity is restored, even acutely during adulthood, this behavioural deficit is improved. These results suggest that *Mecp2* deficiency impairs not only developing neural circuits, but also the function and plasticity of adult circuits, via mechanisms involving PV$^+$ GABAergic networks. GABAergic interneurons are basic components of cortical microcircuits that are conserved across brain areas. The same mechanisms that underlie experience-triggered and MECP2-dependent PV interneuron function during development and adulthood may also apply to other functional modalities affected in RTT.

Emerging evidence indicates that the appropriate expression and function of MECP2 is required in adulthood for normal plasticity and behaviour[8,9]. Remarkably, restoring normal MECP2 expression in adulthood improves behaviour deficits in mice[45,47]. These observations have several implications. First, they indicate that some cellular functions of MECP2 are involved in the maintenance and adult plasticity of neural circuitry, not only its development. Second, they raise the possibility that in humans it may be beneficial to therapeutically restore MECP2 levels at later stages. Nevertheless, the specific mechanisms by which *Mecp2* mutations impair adult neural function need to be elucidated.

Our data demonstrate that heterozygous mutations in *Mecp2* (*Mecp2*^*het*) interfere with auditory cortical plasticity that occurs in adult mice during initial maternal experience. Mothers and wild type virgin surrogates achieve proficiency in pup retrieval behaviour by an experience-dependent learning process[16,19,20,32,48,49], that is correlated with neurophysiological plasticity in the auditory cortex[16,17,18,50]. We used gathering behaviour to assay defects in this sensory plasticity. Our results show that *Mecp2*^*het* have markedly impaired ability to learn appropriate gathering responses to pup calls. This interference is in large part because of a specific requirement for MECP2 in the adult auditory cortex. Deletion of MECP2 in adult mice selectively in the auditory cortex also produced inefficient retrieval. We saw no improvement in the behaviour of the mutants over the first five days post birth. At that point, pups were sufficiently mobile that they no longer required gathering. However, it is tempting to speculate that the *Mecp2*^*het* might improve with more practice, such as with subsequent litters.

Electrophysiological recordings from naive mice of both genotypes demonstrate that there are no gross deficits in basic auditory cortex function in heterozygous mutants and that they are not deaf. We speculate instead that there are more subtle and context-specific impairments of intra-cortical processing and plasticity in the auditory cortex of *Mecp2*^*het*.

We find evidence of dysregulated cortical inhibitory networks during maternal experience in *Mecp2*^*het*. This is consistent with increasing evidence that dysfunction of GABA signalling is associated with autism disorders and RTT (refs 33–35,51). Importantly, disruption of MECP2 in GABAergic neurons recapitulates multiple aspects of RTT including repetitive behaviours and early lethality[34], although the pathogenic mechanisms remain unclear.

Our data suggest that an important aspect of the pathology associated with heterozygous *Mecp2* mutations is impaired plasticity of cortical inhibitory networks. Pup exposure and maternal experience trigger an episode of heightened auditory cortical inhibitory plasticity. For example, GAD67 levels are roughly doubled in the auditory cortex of both wild type and *Mecp2^het* five days after the birth of the litter. This result suggests a reorganization of the cortical GABAergic network triggered by maternal experience. Although this feature of auditory cortex plasticity is shared between SurWT and SurHet, SurHet also show large increases in expression of PV and PNNs on the fifth day of pup exposure. Notably, initial levels of these inhibitory markers in NaiveWT and NaiveHet, and levels in Sur after pups are weaned, are identical. Therefore, potentially crucial features of *Mecp2^het* pathology may only be revealed by the commencement of an episode of heightened sensory and social experience, as occurs with first-time pup exposure. We speculate that this may be a general phenomenon wherein exposure to salient sensory stimuli may define a particularly vulnerable point for *Mecp2^het*. Further assessment using natural stimuli targeting motor and social circuits that challenge network plasticity mechanisms may reveal endo-phenotypes.

Both WT and *Mecp2^het* female mice exhibit low GAD67 expression as maternally-naive adults. Expression sharply increases after exposure to a mother and her pups, and returns to baseline levels when the pups are weaned. This is correlated with a surge in the expression of PV and PNNs in *Mecp2^het* only. This result is consistent with increased PV (ref. 33) and PNN expression observed in the developing *Mecp2*-null visual cortex[39]. Several lines of evidence implicate elevated expression of PV and PNN as brakes that terminate episodes of plasticity in development and adulthood. In the developing cortex, maturation of GABAergic inhibition mediated by the fast-spiking PV interneuron network is a crucial mechanism for regulating the onset and progression of critical periods[36]. During postnatal development, PV interneurons undergo substantial changes in morphology, connectivity, intrinsic and synaptic properties[52–55], and they form extensive reciprocal chemical and electrical synapses[52,56,57]. Learning associated with a range of adult behaviours might rely on similar local circuit mechanism observed in the developing cortex[25,58]. This model is supported by our finding that knockout of *Mecp2* specifically in PV neurons is sufficient to impair pup gathering behaviour. From these results, we speculate that increased PV and PNN expression might support an enhanced inhibitory function that might lead to reduced neuronal activation of excitatory circuits in a stimulus-specific manner, in agreement with previously published reports[30,59,60].

PNNs inhibit adult experience-dependent plasticity in the visual cortex[24], and in consolidating fear memories in the amygdala[61]. PNN assembly in the SurHet tracks with changes in PV expression after maternal experience, suggesting there is remodelling of the extracellular matrix during natural behaviour. This is an interesting observation as the prevailing notion of PNNs during adulthood is as a stable, structural barrier which needs to be removed with chondroitinase ABC to reactivate plasticity. Related to this, there was no further improvement in WT that received ChABC injection possibly revealing a ceiling effect.

We demonstrate that manipulating GAD67 expression using *Gad1* heterozygotes is sufficient to restore normal PV and PNN expression patterns and behaviour. This result suggests a critical role for *Gad1* in regulating MECP2-mediated experience-driven cellular and circuit operations. MECP2 directly occupies the promoter regions of *Gad1* and *PV* (refs 33,34), thus potentially configuring chromatin in these promoter and enhancer regions

for appropriate activity- and experience-dependent regulation. We speculate that MECP2 regulates specific ensembles of genes and the temporal profile of their expression to control the tempo of plasticity. MECP2 regulates many genes[13,62]; therefore there are likely other as yet unappreciated targets that could contribute to this control.

Our data are consistent with an emerging body of literature that suggests that auditory cortical plasticity is triggered in adult female virgin mice by pup exposure. By using pup gathering behaviour as readout of the efficacy of this plasticity, we observe that impaired MECP2 expression disrupts both behaviour and the underlying auditory cortical plasticity. This is consistent with recent data revealing sensory impairments in individuals with RTT, which may contribute to behavioural symptoms[63,64]. We further speculate that MECP2 deficiency results in suppressed ('negative') experience-dependent plasticity[65] that may act at other brain regions and time points to contribute to a range of altered behaviours.

## Methods

**Animals.** All experiments were performed in adult female mice (7–10 weeks old) that were maintained on a 12-h–12-h light-dark cycle (lights on 07:00 h) and received food *ad libitum*. Genotypes used were CBA/CaJ, *Mecp2^het* (C57BL/6 background; B6.129P2(C)-*Mecp2^tm1.1Bird*/J), *Mecp2^wt*, *Mecp2^flox/flox* (B6.129S4-*Mecp2^tm1Jae*/Mmucd) and *PV-ires-Cre* (B6;129P2-Pvalbtm1(cre)Arbr/J). *Mecp2^flox/flox* mice were bred with an H2B-GFP (*Rosa26-loxpSTOPloxp-H2BGFP*) line[66] to facilitate identification of injected cells. The double mutant *Mecp2^het*; *Gad1^het* (Het;*Gad1^het*) was generated by crossing *Mecp2^het* females and *Gad1^het* males. The *Gad1^het* allele was generated using homologous recombination in ES cells; a cassette containing de-stabilized GFP cDNA (D2GFP) was inserted at the translation initiation codon (ATG) of the *Gad1* gene. The goal was to generate a *Gad1* gene transcription reporter allele, but the same allele is also a gene knockout. This design was essentially the same as the widely used *Gad1-GFP* knockin allele[67]. Targeted ES clones were identified by PCR and southern blotting. Positive ES clones were injected into C57BL/6 mice to obtain chimeric mice following standard procedures. Chimeric mice were bred with C57BL/6 mice to obtain germline transmission. D2GFP expression was weak and was restricted to GABAergic neurons throughout the mouse brain, indicating successful gene targeting. The colony is maintained as heterozygotes, as homozygotes are lethal. For genetic knockout of MECP2 in PV cells, we obtained mice heterozygous for *PV-ires-Cre*; *Mecp2^flox/flox* (het-PM) by breeding males homozygous for *PV-ires-Cre* with females homozygous for *MeCP2^flox/flox*. For behavioural and molecular analysis, *het-PM* were bred to obtain females of *PV-ires-Cre^+/−*;*Mecp2^flox/flox+/+* and control littermates (*PV-ires-Cre^−/−*;*Mecp2flox^+/+ or +/−*). All procedures were conducted in accordance with the National Institutes of Health's *Guide for the Care and Use of Laboratory Animals* and approved by the Cold Spring Harbor Laboratory Institutional Animal Care and Use Committee.

**Pup gathering behaviour and movement analysis.** We housed two virgin female mice (one control and one experimental mouse; termed 'surrogates') with a primiparous CBA/CaJ female beginning 1–5 days before birth. Pup retrieval behaviour was assessed starting on the day the pups were born (postnatal day 0; D0) as follows: (1) one female was habituated with 3–5 pups in the nest of the home cage for 5 min, (2) pups were removed from the cage for 2 min and (3) one pup was placed at each corner and one in the center of the home cage (the nest was left empty if there were fewer than 5 pups). Each adult female had maximum of 5 min to gather the pups to the original nest. After testing, all animals and pups were returned to the home cage. The same procedure was performed again at D3 and D5. All behaviours were performed in the dark, during the light cycle (between 10:30 AM and 4:00 PM) and were video recorded. For analysis, an experimenter who was blind to genotype and experimental condition counted the number of errors and measured the latency of each mouse to gather all five pups. An error was scored for each instance of gathering of pups to the wrong location or of interacting with the pups (for example, licking or sniffing) without gathering them to the nest. Normalized latency was calculated using the following formula:

$$\text{latency index} = [(t_1 - t_0) + (t_2 - t_0) + ... + (t_n - t_0)]/(n \times L)$$

where $n =$ # of pups outside the nest, $t_0 =$ start of trial, $t_n =$ time of $n$th pup gathered, $L =$ trial length.

Movement was measured while the animal was performing pup retrieval behaviour, using Matlab-based software (MathWorks)[68].

**Injections.** Mice were anesthetized with ketamine (100 mg kg$^{-1}$) and xylazine (5 mg kg$^{-1}$) and stabilized in a stereotaxic frame. Lesions in the auditory cortex of CBA/CaJ mice were performed by injection of ibotenic acid (0.5 µl of 10 mg ml$^{-1}$

per site; Tocris Bioscience). Control animals were injected with the solvent only (0.9% NaCl solution). Pup retrieval behaviour was evaluated 3–5 days later. To knock down MECP2 expression, we injected AAV9-GFP-IRES-Cre (0.3 μl of $4 \times 10^{12}$ mol ml$^{-1}$ per site; UNC Gene Therapy Center) into the auditory cortex of 4 weeks old $Mecp2^{flox/flox}$ mice. AAV2/7-CMV-EGFP was used as control (both AAV viruses were kind gifts from Dr Bo Li, CSHL). Behaviour was evaluated 4–6 weeks later. To degrade PNNs, we injected chondroitinase ABC (0.3 μl of 50 U ml$^{-1}$ per site, in 0.1% BSA/0.9% NaCl solution; Sigma-Aldrich) into the auditory cortex of $Mecp2^{het}$ and wild type littermate mice. Penicillinase (50 U ml$^{-1}$, in 0.1% BSA/0.9% NaCl solution; Sigma-Aldrich) was used as injection control. Pup retrieval behaviour was evaluated 3–5 days later. For Fig. 7e–h, three ChABC-injected Het mice were excluded from analysis because of mis-targeting of the auditory cortex. The data for these three mice were included in the correlation analysis (Fig. 7i,j). All substances were injected into both auditory cortical hemispheres, two sites per hemisphere, at the following coordinates: bregma $= -2.25$ and $-2.45$ mm, $\sim 4$ mm lateral and 0.75 mm from the dorsal surface of the brain.

**Immunohistochemistry.** Immediately after the behavioural trial on D5, mice were perfused with 4% paraformaldehyde/PBS, and brains were extracted and post-fixed overnight at 4 °C. Brains were then treated with 30% sucrose/PBS overnight at room temperature (RT) and microtome sectioned at 50 μm. Free-floating sections were immunostained using standard protocols at RT. Briefly, sections were blocked in 10% normal goat serum and 1% Triton-X for 2–3 h, and incubated with the following primary antibodies overnight: MECP2 (1:1,000; rabbit; Cell Signaling), PV (1:1,000; mouse; Sigma-Aldrich) and biotin-conjugated Lectin (labels PNNs; 1:500; Sigma-Aldrich). Sections were then incubated with appropriate AlexaFluor dye-conjugated secondary antibodies (1:1,000; Molecular Probes) and mounted in Fluoromount-G (Southern Biotech). To obtain GAD67 staining in soma, three modifications were made according to a previous protocol[69]: (1) no Triton-X or detergent was used in the blocking solution or the antibody diluent; (2) sections were treated with 1% sodium borohydride for 20 min before blocking, to reduce background; and (3) sections were left in GAD67 antibody (mouse; 1:1,000; Millipore) for 48–60 h at room temperature. Brains of all uninjected mice were processed together with the mothers at all steps in the process (perfusions, sectioning, immunostaining and imaging with the same settings). Brains of injected mice were processed together with their respective controls at all steps. Brain sections for MECP2 expression analysis (Fig. 1h,i) and from MECP2 knockdown experiment were further counterstained with a nuclear marker, DAPI.

**Image acquisition and analysis.** To determine the percentage of cell population expressing MECP2 (Fig. 1h,i), all DAPI$^+$ whole cells within a region of interest (100 μm × 100 μm) in the ×20 projection image were determined to be either positive or negative for MECP2 expression. Percentage was calculated by dividing the number of DAPI$^+$ cells with MECP2 expression by the total number of DAPI$^+$ cells. Each data point in Fig. 1h,i, represents an average percentage value calculated from four ×20 projection images for each mouse.

To analyse percentage infection of the auditory cortex by AAV-GFP-Cre or degradation of PNNs by chondroitinase ABC, 4–5 single-plane images per auditory cortical hemisphere from each animal were acquired using Olympus BX43 microscope (×4 objective, UPlanFL N) and analysed using ImageJ (NIH). To calculate percentage infection/degradation in each image, the area of the entire auditory cortex was measured based on Allen brain atlas boundaries (Version 1, 2008). Then, the area containing GFP$^+$ cells or reduced PNN expression was measured and divided by the total auditory cortical area. For non-auditory cortical region analysis, cumulative regions included temporal association cortex, entorhinal cortex and perirhinal cortex. Each correlation data point represents the percentage infection/degradation per animal.

To determine the percentage of AAV-GFP-Cre infected cells lacking MECP2 expression, four confocal images of the auditory cortex (two images per hemisphere) were acquired using the Zeiss LSM710 confocal microscope (×20 objective; ×2 zoom) for each AAV-GFP-Cre injected mouse. Using ImageJ (NIH), a region of 100 μm$^2$ was used to determine the percentage of GFP$^+$ cells that lack MECP2 expression.

For Fig. 2f, the amount of MECP2 knockdown was assessed by comparing MECP2 intensity in infected cells (GFP$^+$) and uninfected cells (GFP$^-$) within the same auditory cortical region of each AAV-GFP-Cre injected mouse. 2 confocal images of the auditory cortex (1 image per hemisphere) were acquired using the Zeiss LSM710 confocal microscope (×20 objective; ×1 zoom) for each mouse. Using ImageJ and a region of 150 μm$^2$ from each confocal image, the intensity of MECP2 for each GFP$^+$ infected cell was obtained and compared with the intensity of MECP2 in MECP2$^+$ cells that lack GFP (uninfected). Only cells with their entire soma visible in the confocal images were used for the analysis.

To analyse GAD67$^+$ and PV$^+$ soma and PNNs, two confocal images from each auditory cortical hemisphere of each animal were acquired using the Zeiss LSM710 confocal microscope (×20 objective; ×0.6 zoom) and analysed using the LSM Image Browser. Each confocal image of the same hemisphere was separated by at least 150 μm to minimize the counting of the same cells. Scans from each channel were collected in the multiple-track mode. Maximum intensity projections of the Z-stacks were obtained using the 'Projection' setting in the Zeiss LSM Image

Browser. To count high-intensity GAD67$^+$ soma and mature PNNs, the 'Contrast' setting in the Browser was set to 100 to threshold weaker signals. All GAD67$^+$ soma and mature PNNs within the projection images were counted manually. Measurement of PV$^+$ cell intensity was performed using Volocity (Perkin Elmer). PV confocal images were first merged. Then, cell identity and intensity were measured using the option 'Find 2D nuclei' with 'separate touching nuclei = 5 μm' and 'reject nuclei of area < 10 μm$^2$.' Results were confirmed manually to exclude non-cell objects and to include any missed PV$^+$ cells. Finally, obtained cell intensities were background subtracted. The experimenter performing the analysis was blinded to all genotypes and conditions. All statistical analysis was performed using Origin Pro (Origin Lab) and Matlab (MathWorks). All graphs were generated using GraphPad Prism (GraphPad Software). Data are represented as mean ± s.e.m.

**In vivo physiology.** For awake-state recordings, we anesthetized $Mecp2^{het}$ mice and $Mecp2^{wt}$ mice with an 80:20 mixture (1.00 ml kg$^{-1}$) of ketamine (100 mg ml$^{-1}$) and xylazine (20 mg ml$^{-1}$) (KX) and stabilized in a stereotaxic frame. A head bar was affixed above the cerebellum with RelyX Luting Cement (3M) and methyl methacrylate-based dental cement (TEETS). For additional support, five machine screws (Amazon Supply) were placed in the skull before cement application. After one day of recovery, mice were anesthetized with iso-flurane (Fluriso; Vet One) and small craniotomies were made to expose the left hemisphere of auditory cortex. Mice were then head-fixed via the attached head bar over a foam wheel that was suspended above the air table. The foam wheel allowed mice to walk and run in one dimension (forward-reverse).

Stimuli were presented via ED1 Electrostatic Speaker Driver and ES1 Electrostatic Speaker (TDT), in a sound attenuation chamber (Industrial Acoustics) at 65 dB SPL RMS measured at the animal's head. Stimuli consisted of 100-ms presentation of broadband noise, four logarithmically-spaced tones ranging between 4 and 32 kHz, and ultrasound noise bandpassed between 40 and 60 kHz.

Single units were blindly recorded in vivo by the loose-patch technique using borosilicate glass micropipettes (7–30 MΩ) tip-filled with intracellular solution (mM: 125 potassium gluconate, 10 potassium chloride, 2 magnesium chloride and 10 HEPES, pH 7.2). Spike activity was recorded using BA-03X bridge amplifier (npi Electronic Instruments), low-pass filtered at 3 kHz and digitized at 10 kHz, and acquired using Spike2 (Cambridge Electronic Design). Data were analysed using Spike2 and Matlab.

Baseline spontaneous activity was calculated using a 2-second window taken before the onset of stimuli. To assess statistical significance of responses to individual stimulus, we used a bootstrap procedure as follows. If $n$ trials were collected with the response window length $t$ (100 ms), then a distribution was created by sampling $n$ length $t$ windows from the full spike record 10,000 times and taking the mean deviation of each window from the spike rate measured in the prior 2 s. Responses that were in the top or bottom 2.5% of this distribution were deemed significantly excitatory or inhibitory, respectively.

**Data availability.** The data that support the findings of this study are available from the corresponding author on reasonable request.

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

## Acknowledgements

We wish to thank D. Huang and A. Chandrasekhar for data collection and analysis assistance, Alexandra Nowlan for help with the mouse drawing and Stephen Hearn at the CSHL microscopy facility for assistance with Volocity software. We would also like to thank A. Zador, B. Li, R. Froemke, J. Tollkuhn, J. Morgan, D. Eckmeier, B. Cazakoff, A. Fleischmann and A. Maffei for helpful comments and discussion. This work was supported by grants to S.D.S. from the Simons Foundation Autism Research Initiative

(SFARI) and the National Institute of Mental Health (R01MH106656), to ZJH from the National Institute of Mental Health (RO1MH102616) and to KK from National Alliance for Research on Schizophrenia and Depression Young Investigator Grant from the Brain and Behaviour Research Foundation and an International Rett Syndrome Foundation Postdoctoral Fellowship.

## Author contributions

S.D.S. and Z.J.H. supervised the project. K.K., B.Y.B.L. and S.D.S. designed the experiments and developed the methods. K.K., B.Y.B.L., G.E. and S.D.S. collected and analysed the data. K.K., B.Y.B.L., Z.J.H. and S.D.S. wrote and edited the paper.

## Additional information

**Competing financial interests:** The authors declare no competing financial interests.

