## [Peer Review File · Nature Communications]

Reviewers' comments:

Reviewer #1 (Remarks to the Author):

This study by Krishnan et al. investigates roles for MeCP2 in adult plasticity upon learning. They use a pup retrieval assay in virgin wildtype and MeCP2het females that learn this maternal behaviour by being housed together with a mother and her pups. Previous studies by other labs have shown that this learning depends on auditory cortex plasticity involved in detecting distress ultrasounds by pups. MeCP2het fail to learn this task, and wild type mice with a conditional deletion of MeCP2 in auditory cortex exhibit a comparable phenotype. At the cellular level, learning involves increased expression of GAD67 in auditory cortex interneurons in wild type and mutant mice, but the mutants further exhibit strongly elevated PV expression and perineuronal net assembly. These responses in wild type and mutant mice are specifically detected during learning (Day5) and reverse to baseline upon weaning (Day21). Crossing into a GAD1+/- background rescues the PV phenotype and learning, a finding consistent with previous reports that GAD1 (i.e. GAD67) promotes PV expression.

The authors conclude that MeCP2het females exhibit a dysregulation in learning-induced interneuron plasticity in the adult, consisting in overexpression of PV and of PNNs specifically upon learning, leading to pronounced learning failure.

These are interesting findings. By combining the pup retrieval assay to MeCP2 deficiency the authors provide an attractive setting to investigate roles of MeCP2 in adult plasticity and learning. The study makes a convincing case for excess PV and PNN induction in a learning setting involving GAD67 induction (i.e. interneuron plasticity). GAD67 induction upon learning is not affected in the mutants (but is it excessive?), suggesting that aspects of PV neuron plasticity are specifically dysregulated (consistent with excessive plasticity), leading to a failure in learning.

Unfortunately, however, the advance over previous knowledge is modest. The same authors had shown in a PNAS study in 2015 that lack of MeCP2 interferes with critical period maturation in visual cortex - a process known to be critically dependent on PV neuron plasticity. The pup retrieval assay is not new, and it was known that it depends on plasticity in auditory cortex. Furthermore, previous studies have already related MeCP2 deficits to alterations in GABAergic neuron function. In particular, a study by He LJ et al. (not cited here) in Nat. Commun. (2014) provided evidence that conditional deletion of MeCP2 in PV neurons interferes with critical period plasticity.

Overall, this study is of potential interest, but in order to provide a sufficient advance over previous studies it would need to more precisely (and if possible mechanistically) relate lack of MeCP2 to abnormal PV neuron plasticity upon learning. In the absence of firm mechanistic advances, which would be ideal, the study might be strengthened by additional experiments focusing on PV neurons, and providing a more precise account of how their learning-induced plasticity is specifically affected in MeCP2het. A further possibility might involve conditional deletion of MeCP2 in auditory cortex PV neurons.

Additional point:

The discussion section mainly repeats the main findings, and relates them to conclusions from previous studies. The discussion would be more informative if it would focus on what is novel here, on what specific aspects of interneuron plasticity are altered in MeCP2het, and on possible mechanisms relating MeCP2 deficiency to PV neuron plasticity.

Reviewer #2 (Remarks to the Author):

In the manuscript by Krishnan, et al., the authors present evidence that a form of auditory

learning in adult female mice is impaired in a mouse model of Rett Syndrome (Mecp2het). The authors demonstrate that markers of inhibitory synaptic transmission and extracellular matrix components (PNNs) are increased in auditory cortex as part of the normal response to cohabitation with a mother and pups and that these changes are enhanced in the Mecp2het mice. Importantly, the authors found that the deficits could be reversed by manipulations of the GABAergic system, making for a compelling study. The manuscript is generally well written and the data clearly presented. The study is important on several levels, from both building on the authors' previous reported findings on PNNs in Mecp2 mutant mice and novel findings on auditory learning, to serving as a cautionary example for other investigators using these (and other) mice in their rearing abilities. The few important issues needing to be addressed are mostly related to reporting and statistics. They are listed below along with other minor issues that should be addressed.

- 1) Throughout the manuscript there are several issues with reporting of statistics. For example, with ANOVAs, the authors will need to report main effect sizes and comparisons, in addition to the post-hoc p values. As an aside, the authors will need to standardize their nomenclature: *P, p, #P are all used within the text. Use capital N for number of animals. Small n typically refers to number of replications (as in cells).
- 2) Reporting the number of images used in quantification is not particularly helpful, in that the reader would need to go to the methods to determine magnification and other parameters such as number and thickness of sections per animal. More useful would be to present the number of cells counted per animal or per group.
- 3) The authors make no mention of whether Mecp2het brains/cells are smaller (like in Mecp2 KO mice), and how any effect on cell size would affect the quantification. Everything should be normalized to area instead of raw cell counts (or done in addition as supplemental analyses).
- 4) Along the same lines, more information is needed on the cell quantification to determine whether it was performed in a rigorous and unbiased manner, such as a) whether nth serial sections were taken to minimize counting of the same cells, b) were all the cells in an image counted, and if not, what was the criteria for inclusion?
- 5) In many instances units are not provided with statistics or in figure legends (i.e. latency be reported in a unit of time)
- 6) in vivo physiology is reported for naïve hets only. The authors will need to be able to compare these to WT numbers.
- 7) I appreciate that the individual animal data points for behavior are presented. This should be used throughout though (they are missing in fig 6).
- 8) Bottom of page 7/ top of 8- Although it is useful in establishing that the NaiveHet mice are not deaf by quantifying the mean number of stimuli that evoked a significant response in auditory neurons, it is unclear how this compares to the number in wild-type. Was this done? If not, the authors will need to do that comparison or cite relevant work so readers can determine whether the mice have 'normal' hearing.
- 9) Figure 5- 5C: did the PV population of SurGad1het shift to WT numbers for both high and low PV expression level-cells? Please clarify. 5F/G: Were experiments performed to show latency and error behavior of Naïve Gad1het versus Naïve MECP2het; Gad1het? These would be informative controls.

Minor points:

-Use correct nomenclature for mouse genes (Mecp2 not MeCP2)

-The term "retrieval learning" often refers to a completely different memory process that may be confusing to some readers. Possibly use "pup gathering" as mentioned in other places throughout the text or other phrase consistently to avoid this confusion.

-Do the Mecp2 het dams have increased PNNs compared to WT outside of auditory cortex as well? Please state (or provide a reference to that effect).

- Figure 4- the graphs, especially those in B and C are confusing. An additional legend on the graph would be helpful to clarify what each color refers to or use one color to refer to all 'low PV intensity neurons'. Also the Y-axis of 4C would be more accurately described as "# PNN+ cells/ volume" according to the methods. (Also in Figure 5)

-The abbreviations designated for groups, such as Sur and Naïve, are not consistently used throughout the text, often reverting to the full word. This should stay consistent and be applied to both WT and HET genotypes to minimize confusion. For example Gad1het and NaïveHet;Gad1het are used. Should the prior be NaïveWT;Gad1het? Also, Naive vs Naïve used interchangeably and should instead be consistent (but likely per Journal style).

-Cream- and yellow-colored lines in Figures 4 and 5 (NaïveHet) are not visible.

-Figure 6: please show an example of the extent of chondroitinase effect using a lower magnification image. Again, it is not true that the number of PNNs is shown- it is the number of PNN+ cells. Because this type of counting necessarily would miss the modestly-labeled cells, a more accurate measure would be to simply express the data as relative fluorescence levels.

-Report (within the text) that injections were bilateral.

-Considering that injection of the chondroitinase did not influence the latency or errors of pup retrieval in WT animals, despite the noted decrease in PNNs, a discussion on these findings would be useful.

Reviewer #3 (Remarks to the Author):

De novo mutations of the methyl binding protein MeCP2 underlie the onset of Rett Syndrome, a neurodevelopmental disorders characterized by regression after initial apparent normal development. Disruption in wiring and function of inhibitory circuits affect experience-dependent plasticity in early postnatal life and significantly contributes to the onset and progression of the disorder. In the present manuscript, Krishnan et al. analyzed the function of MeCP2 in adult cortical plasticity at the circuit and behavioral levels. The authors designed a very clever paradigm of experience-dependent behavior of pup retrieval and compared adult MeCP2 heterozygote (MeCP2Het) to WT females exposed for the first time to a "normal" mother with pups. First they established the retrieval paradigm in surrogates WT females and showed that the auditory cortex is necessary for the execution of such task. They then went on and demonstrated that surrogates MeCP2Het exhibit a defect in the gathering behavior. At the circuit level, they discovered that the retrieval paradigm induces a transient yet significant increase in the expression of GABA synthesizing enzyme GAD67. Interestingly, MeCP2Het mice displayed a further up-regulation of parvalbumin (PV) and perineuronal nets (PNN) expression after exposure to pups. Interestingly, genetic down-regulation of GAD67 or pharmacological destruction of PNN was sufficient to renormalize PV expression and rescue behavioral defects. The authors then concluded that Mecp2 regulates this experience-dependent plasticity by affecting PV positive inhibitory circuit maturation and function. The work is well-done, the data are robust and the results are very relevant and novel. The authors need to address only few points before the manuscript can be suitable for publication in Nat Comm.

1) It is well known how difficult is to maintain Mecp2-deficient colonies due to the combination of poor maternal care and reduced vocalization of Mecp2 deficient pups. However, MeCP2Het mice eventually become better mothers with age and successfully take care of their pups. This raises the possibility of a slower learning curve for maternal behavior possibly due to reduced experience-dependent plasticity as shown here or delayed maturation. The authors should address this point in the discussion.

2) It is well know that the onset of Rett Syndrome phenotype in MeCP2Het females is very variable

(due to the X-inactivation) and directly correlate with the cortical phenotype (LeBlanc et al., 2015). What was the phenotypic score of the MeCP2Het at the beginning and end of the learning paradigm? Did the score correlate with the severity of impairment in the gathering behavior?

3) Auditory-evoked neural responses are markedly and significantly reduced in MeCP2 mutants compared to WT animals (Goffin et al., 2012; 2014) indicating that, although mice are not deaf, their ability to perceive and process auditory inputs may be compromised. The in vivo electrophysiological recordings of auditory responses to pure tones and noise are therefore very relevant and as such the data should be presented in a more extensive format. Specifically the authors should plot the spontaneous and evoked activity found in the MeCP2Het mice and compared to WT animals. Ideally, same recordings should be performed after the pup retrieval training. The altered expression of inhibitory markers and PNN found in the MeCP2Het support an enhanced inhibitory function that may explained a reduced neuronal activation of excitatory circuits. These points and the relative literature (Goffin et al., 2012; 2014 but also Durand et al., Kron et al., 2012; Patrizi et al., 2015) should be discussed and appropriately cited.

Reviewer one

The reviewer begins by offering some praise for our study, stating that we make "a convincing case for excess PV and PNN induction in a learning setting involving GAD67 induction (i.e. interneuron plasticity)." However, the reviewer also feels that this is not a significant advance over other recent studies, including one that shares several authors with our manuscript:

Unfortunately, however, the advance over previous knowledge is modest. The same authors had shown in a PNAS study in 2015 that lack of MeCP2 interferes with critical period maturation in visual cortex - a process known to be critically dependent on PV neuron plasticity. The pup retrieval assay is not new, and it was known that it depends on plasticity in auditory cortex. Furthermore, previous studies have already related MeCP2 deficits to alterations in GABAergic neuron function. In particular, a study by He LJ et al. (not cited here) in Nat. Commun. (2014) provided evidence that conditional deletion of MeCP2 in PV neurons interferes with critical period plasticity.

We disagree with this assessment, and we note that the reviewer seems unaware of critical differences between our study and the two that they mention.

1) The previous studies were performed in juvenile mice undergoing a developmentally programmed critical period. Our study was performed in adult mice undergoing an episode of experientially-triggered cortical plasticity. This is a crucial point because it suggests that *Mecp2* mutation impairs not only developmental critical periods, but also adult learning, and that it does so through a similar mechanism. Indeed, this is the core finding of our work, and it is novel.

2) A corollary of this finding is that this form of adult learning in wild type mice seems to access mechanisms shared with developmental critical periods (e.g. PV interneuron plasticity). Importantly, these mechanisms are initiated by experience, not a developmental program. This is also a key finding of our study and we are unaware of any precedent for this concept in the literature.

3) The previous studies were performed in male null mutants. Although it is an implicit assumption of studies performed in male nulls, it is unclear that the pathology of those mice is reflective of the circuit impairment in heterozygous mutants. Because *Mecp2* is on the X chromosome, heterozygous mice are mosaic for its expression. In other words, individual cells

either express MECP2 normally or not at all. The mosaic brain may exhibit properties that are very different from those of the null brain. Humans with Rett syndrome are heterozygous females, so our model is a more accurate genocopy of the human condition. Although other studies have used these mice, all previous studies of critical period plasticity have used null males.

4) One of the most unique aspects of our study is the inclusion of a complex and natural behavioral measure that apparently closely reflects cortical plasticity. Indeed, as the reviewer points out it is not new, but its application is a functional assay for plasticity deficiencies in mouse mutants is entirely new. Previously the behavior was correlated with plasticity of field potentials and spiking responses of unidentified auditory cortical neurons (Gallindo-Leon, 2010). Based on the field potentials, the authors argued that the plasticity involved changes in inhibition, however the nature of the data left many questions regarding the circuit implementation of those changes. A more recent study (Marlin et al, 2015) implicated oxytocin release in changing the balance of excitation and inhibition in the cortex during retrieval learning. These studies and others that support the link between auditory cortical plasticity and maternal retrieval learning are what make this behavior a strong candidate for a novel functional assay of adult cortical plasticity in mouse models of ASD. This past work is a strength, not a weakness. Finally, even with all of this past work, our study is the first to make any perturbation that was specific to the auditory cortex that impaired pup retrieval.

Overall, this study is of potential interest, but in order to provide a sufficient advance over previous studies it would need to more precisely (and if possible mechanistically) relate lack of MeCP2 to abnormal PV neuron plasticity upon learning. In the absence of firm mechanistic advances, which would be ideal, the study might be strengthened by additional experiments focusing on PV neurons, and providing a more precise account of how their learning-induced plasticity is specifically affected in MeCP2het. A further possibility might involve conditional deletion of MeCP2 in auditory cortex PV neurons.

As we discussed above, our study is already quite novel in many respects that the reviewer does not acknowledge. Nevertheless, to address their concern we have now performed the reviewer's suggested experiment of knocking out *Mecp2* selectively in PV cells. The data clearly show that this selective manipulation is sufficient to significantly impair retrieval behavior in early trials, as shown in Figure 8. The data are also reported in the text on pp. 16. Interestingly, these

mice are able to compensate for this deficiency in later trials, suggesting a role for PV cells in the initiation but not consolidation of plasticity. Consistent with the recovery, we now report that mice that experienced PV cell deletion of *Mecp2* exhibited low PNN expression at day five. Therefore, this raises the possibility that sustained pathology and behavioral disruption are achieved by elevated PNN expression. These data strengthen the conclusion that PV-expressing neurons have an important role in the disruption of cortical plasticity in *Mecp2*^{het}.

The discussion section mainly repeats the main findings, and relates them to conclusions from previous studies. The discussion would be more informative if it would focus on what is novel here, on what specific aspects of interneuron plasticity are altered in MeCP2het, and on possible mechanisms relating MeCP2 deficiency to PV neuron plasticity.

We have made changes to the discussion (pp. 17-18) that reflect the points of novelty mentioned above, include discussion of the new data from PV cell knockouts, and speculate more regarding potential mechanisms that explain our results (p. 21).

We wish to thank reviewer one very much for their review and we sincerely hope that we have addressed their concerns.

Reviewer two

Many thanks to reviewer two for their thoughtful and detailed critique.

*1) Throughout the manuscript there are several issues with reporting of statistics. For example, with ANOVAs, the authors will need to report main effect sizes and comparisons, in addition to the post-hoc p values. As an aside, the authors will need to standardize their nomenclature: *P, p, #P are all used within the text. Use capital N for number of animals. Small n typically refers to number of replications (as in cells).*

We thank the reviewer for bringing these oversights to our attention. They have all been rectified.

2) Reporting the number of images used in quantification is not particularly helpful, in that the reader would need to go to the methods to determine magnification and other

parameters such as number and thickness of sections per animal. More useful would be to present the number of cells counted per animal or per group.

This has been updated accordingly.

3) The authors make no mention of whether Mecp2het brains/cells are smaller (like in Mecp2 KO mice), and how any effect on cell size would affect the quantification. Everything should be normalized to area instead of raw cell counts (or done in addition as supplemental analyses).

We now report cell counts normalized to volume.

By gross measures, no significant difference in brain or cell size has been observed in *Mecp2^{het}*. A recent paper reported a mild 5% decrease in nuclear volume of MECP2-negative cells in *Mecp2^{het}* by a sophisticated high-resolution imaging approach (Linhoff, Garg and Mandel, Cell, 2015, 163(1):246).

In our assay, cell size is difficult to measure because it can be confounded with the intensity changes we saw in the markers we used. Because we used discrete cell counts for all markers, we do not think our results can be due to changes in cell size.

4) Along the same lines, more information is needed on the cell quantification to determine whether it was performed in a rigorous and unbiased manner, such as a) whether nth serial sections were taken to minimize counting of the same cells, b) were all the cells in an image counted, and if not, what was the criteria for inclusion?

We have added many details to the reporting that clarify these issues in Methods (pp. 28-29).

5) In many instances units are not provided with statistics or in figure legends (i.e. latency be reported in a unit of time)

Our measure of latency is a normalized measure computed by the equation in the methods and ranges from 0 to 1. It is not a time unit. It is the fraction of the maximum time a mouse could take to perform the task, given the number of pups. To clarify this, we now refer to the measure

as 'latency index' to reinforce this distinction.

6) in vivo physiology is reported for naïve hets only. The authors will need to be able to compare these to WT numbers.

We added these data. They can be found in Figure 2 and are reported in the text on pp. 8.

7) I appreciate that the individual animal data points for behavior are presented. This should be used throughout though (they are missing in fig 6).

The former Figure 6 (now Figure 7 E,F) depicts three groups of mice at three time points, so it is not practical to show all the data points. Thus we added panels G and H to Figure 7 that show all data points for Day 5. That is the day that ChABC injected animals exhibit maximum improvement.

8) Bottom of page 7/ top of 8- Although it is useful in establishing that the NaiveHet mice are not deaf by quantifying the mean number of stimuli that evoked a significant response in auditory neurons, it is unclear how this compares to the number in wild-type. Was this done? If not, the authors will need to do that comparison or cite relevant work so readers can determine whether the mice have 'normal' hearing.

As we mentioned above in response to point 6, these data are now included in Figure 2. We also cite Goffin et al, 2014 paper, which shows intact MECP2 is not required for 'normal' hearing.

9) Figure 5- 5C: did the PV population of SurGad1het shift to WT numbers for both high and low PV expression level-cells? Please clarify. 5F/G: Were experiments performed to show latency and error behavior of Naïve Gad1het versus Naïve MECP2het; Gad1het? These would be informative controls.

We apologize that the first point was not clear in the original submission. We now explicitly state that there is no significant difference in PV intensity between WT surrogates and surrogate *WT;Gad1^{het}* on page 13. With regard to the second point, we did not assess naïve behavior in any genotype other than wild type, including *Mecp2^{het}*. We found poor and variable naïve

behavior even in WT, as shown in Figure 1B. It seems unlikely that the mutants would be any better, so we don't think these experiments would be very sensitive or informative.

Minor points:

-Use correct nomenclature for mouse genes (Mecp2 not MeCP2)

This has been fixed.

-The term "retrieval learning" often refers to a completely different memory process that may be confusing to some readers. Possibly use "pup gathering" as mentioned in other places throughout the text or other phrase consistently to avoid this confusion.

Great point. We now avoid the term.

-Do the Mecp2 het dams have increased PNNs compared to WT outside of auditory cortex as well? Please state (or provide a reference to that effect).

There are no reports in the literature of differences in PNN intensity between *Mecp2^{het}* and WT. We did not measure changes in other regions. We chose to focus on the auditory cortex because our data show that selective manipulations of this structure are necessary (*Mecp2-flox* experiments) and sufficient (ChABC experiments) for the behavior. Therefore, we reason that even if there are changes elsewhere, they don't bear on this behavior.

*- Figure 4- the graphs, especially those in B and C are confusing. An additional legend on the graph would be helpful to clarify what each color refers to or use one color to refer to **all** 'low PV intensity neurons'. Also the Y-axis of 4C would be more accurately described as "# PNN+ cells/ volume" according to the methods. (Also in Figure 5)*

*-The abbreviations designated for groups, such as Sur and Naïve, are not consistently used throughout the text, often reverting to the full word. This should stay consistent and be applied to both WT and HET genotypes to minimize confusion. For example *Gad1het* and *NaïveHet;Gad1het* are used. Should the prior be *NaïveWT;Gad1het*? Also, *Naive* vs *Naïve* used interchangeably and should instead be consistent (but likely per Journal style).*

-Cream- and yellow-colored lines in Figures 4 and 5 (NaïveHet) are not visible.

All of these issues have been fixed.

-Figure 6: please show an example of the extent of chondroitinase effect using a lower magnification image. Again, it is not true that the number of PNNs is shown- it is the number of PNN+ cells. Because this type of counting necessarily would miss the modestly-labeled cells, a more accurate measure would be to simply express the data as relative fluorescence levels.

Figure 7 A, B now show low magnification images of the effects of ChABC on PNN intensity. Because PNNs are spatially condensed in our images, there are many unlabeled pixels that would make overall pixel intensity an insensitive measure of the changes after pup experience.

-Report (within the text) that injections were bilateral.

Done.

-Considering that injection of the chondroitinase did not influence the latency or errors of pup retrieval in WT animals, despite the noted decrease in PNNs, a discussion on these findings would be useful.

We have added some discussion of this point on p. 21.

Reviewer three

De novo mutations of the methyl binding protein MeCP2 underlie the onset of Rett Syndrome, a neurodevelopmental disorders characterized by regression after initial apparent normal development. Disruption in wiring and function of inhibitory circuits affect experience-dependent plasticity in early postnatal life and significantly contributes to the onset and progression of the disorder. In the present manuscript, Krishnan et al. analyzed the function of MeCP2 in adult cortical plasticity at the circuit and behavioral levels. The authors designed a very clever paradigm of experience-dependent behavior of pup

retrieval and compared adult MeCP2 heterozygote (MeCP2Het) to WT females exposed for the first time to a "normal" mother with pups. First they established the retrieval paradigm in surrogates WT females and showed that the auditory cortex is necessary for the execution of such task. They then went on and demonstrated that surrogates MeCP2Het exhibit a defect in the gathering behavior. At the circuit level, they discovered that the retrieval paradigm induces a transient yet significant increase in the expression of GABA synthesizing enzyme GAD67. Interestingly, MeCP2Het mice displayed a further up-regulation of parvalbumin (PV) and perineuronal nets (PNN) expression after exposure to pups. Interestingly, genetic down-regulation of GAD67 or pharmacological destruction of PNN was sufficient to renormalize PV expression and rescue behavioral defects. The authors then concluded that Mecp2 regulates this experience-dependent plasticity by affecting PV positive inhibitory circuit maturation and function. The work is well-done, the data are robust and the results are very relevant and novel. The authors need to address only few points before the manuscript can be suitable for publication in Nat Comm.

Thank you to reviewer three for their insightful and complementary remarks.

- 1) It is well known how difficult is to maintain Mecp2-deficient colonies due to the combination of poor maternal care and reduced vocalization of Mecp2 deficient pups. However, MeCP2Het mice eventually become better mothers with age and successfully take care of their pups. This raises the possibility of a slower learning curve for maternal behavior possibly due to reduced experience-dependent plasticity as shown here or delayed maturation. The authors should address this point in the discussion.*

This is an excellent point and one that we agree deserves discussion. We have added material to the discussion section that covers this (pp. 18-19).

- 2) It is well know that the onset of Rett Syndrome phenotype in MeCP2Het females is very variable (due to the X-inactivation) and directly correlate with the cortical phenotype (LeBlanc et al., 2015). What was the phenotypic score of the MeCP2Het at the beginning and end of the learning paradigm? Did the score correlate with the severity of impairment in the gathering behavior?*

LeBlanc et al (Ann Neurol, 2015, 78(5):775) assessed the phenotype scores of mice aged 5 – 64 weeks and reported that older mice in that range had more severe phenotypes. Phenotypic score was correlated with impaired visual evoked potential (VEP) responses. Mice in our study are at the young end of LeBlanc et al's range (7-10 weeks old), and are unlikely to score poorly. We did not formally score our mice; however, we looked for typical phenotypic characteristics including impaired gait and mobility, limb claspings, tremor or other negative conditions during experimentation, and did not find detectable deficits.

Indeed we think variability in X-inactivation affects pup retrieval behavior. We now report that the performance on the task is significantly positively correlated with the number of neurons that express *Mecp2*. These new data are presented in Figure 1 H,I and are reported in the text on p. 7.

3) Auditory-evoked neural responses are markedly and significantly reduced in MeCP2 mutants compared to WT animals (Goffin et al., 2012; 2014) indicating that, although mice are not deaf, their ability to perceive and process auditory inputs may be compromised. The in vivo electrophysiological recordings of auditory responses to pure tones and noise are therefore very relevant and as such the data should be presented in a more extensive format. Specifically the authors should plot the spontaneous and evoked activity found in the MeCP2Het mice and compared to WT animals. Ideally, same recordings should be performed after the pup retrieval training. The altered expression of inhibitory markers and PNN found in the MeCP2Het support an enhanced inhibitory function that may explained a reduced neuronal activation of excitatory circuits. These points and the relative literature (Goffin et al., 2012; 2014 but also Durand et al., Kron et al., 2012; Patrizi et al., 2015) should be discussed and appropriately cited.

The reviewer makes several good points, which are in agreement with our data and interpretation. Our sole objective in presenting the physiology data was to provide a control experiment to show that the mice aren't deaf, as shown in Goffin et al, 2014. The second, and more important point, is that just because the mice aren't deaf doesn't mean there's nothing wrong with their auditory perception. Indeed, that is exactly what we think is wrong with them. Importantly, as the reviewer suggests, there seem to be problems with inhibition that emerge specifically during pup experience.

We now report baseline spontaneous activity is indistinguishable between WT and *Mecp2^{het}*, and that evoked responses show minor differences with different measures. These results have prompted us to collect a large and complex data set with naïve and surrogates. As suggested, we have added these points to the discussion section.

We hope the reviewer finds our expanded reporting of the naïve data satisfying. Relevant experience is a key factor in revealing the crucial features of *Mecp2-het* pathology, as we show in this manuscript. We are exploring this further using *in vivo* physiology and the data analysis extends far beyond the reasonable scope for this manuscript.

REVIEWERS' COMMENTS:

Reviewer #1 (Remarks to the Author):

The authors have addressed several of the points raised by the reviewers and the manuscript can now be recommended for publication in Nat. Communications.

I would like to point out that relating PV neuron plasticity to learning in the adult is definitely NOT novel - this in spite of the statements by the authors in their rebuttal. Nevertheless, the MeCP2 data are of sufficient mechanistic interest.

Reviewer #3 (Remarks to the Author):

The authors have substantially revised their work and the major concerns raised by reviewers have been addressed. Only minor points still need attention before the manuscript can be suitable for publication in Nature Communications:

- 1) Figure legend 2 D, E: it is incorrectly states that "Response strength, as determined by z score, for excitation (D) and inhibition (E) were not significantly different between NaiveWT and NaiveHet...". Indeed response strength for inhibition was significantly increased ($p=0.0054$) as described in main text. Please correct.
- 2) In figure 5 double labeling of PNN/PV should be presented and the quantification reported.
- 3) The images in Figures 4, 5 and 7 should indicate the cortical layers

REVIEWERS' COMMENTS

Reviewer #1 (Remarks to the Author):

The authors have addressed several of the points raised by the reviewers and the manuscript can now be recommended for publication in Nat. Communications.

I would like to point out that relating PV neuron plasticity to learning in the adult is definitely NOT novel - this in spite of the statements by the authors in their rebuttal. Nevertheless, the MeCP2 data are of sufficient mechanistic interest.

Indeed, PV neuron plasticity has been shown to associate with learning (Donato et al., 2013), and we acknowledge this in the manuscript. We are pleased that the reviewer finds our *Mecp2* data to be of mechanistic interest, and we agree.

Reviewer #3 (Remarks to the Author):

The authors have substantially revised their work and the major concerns raised by reviewers have been addressed. Only minor points still need attention before the manuscript can be suitable for publication in Nature Communications:

1) Figure legend 2 D, E: it is incorrectly states that "Response strength, as determined by z score, for excitation (D) and inhibition (E) were not significantly different between NaiveWT and NaiveHet...". Indeed response strength for inhibition was significantly increased ($p=0.0054$) as described in main text. Please correct.

Thank you for pointing out this mistake. It has been corrected.

2) In figure 5 double labeling of PNN/PV should be presented and the quantification reported.

We have now added this data in Figure 5 D, E and H.

3) The images in Figures 4, 5 and 7 should indicate the cortical layers

Cortical layers have been specified now.